# Exploring the Upper Limits of Text-Based Collaborative Filtering Using LLMs: Discoveries and Insights

## Abstract

Text-based collaborative filtering (TCF) has become the mainstream approach for text and news recommendation, utilizing text encoders, commonly referred to as language models (LMs), to represent items. However, the current landscape of TCF models predominantly revolves around the utilization of small or medium-sized LMs. It remains uncertain what impact replacing the item encoder with one of the largest and most potent LMs (LLMs for short), such as the 175-billion parameter GPT-3 model (Brown et al., 2020), would have on recommendation performance. Can we expect unprecedented results? To this end, we conduct an extensive series of experiments aimed at exploring the performance limits of the TCF paradigm. Specifically, we progressively increase the size of item encoders from one hundred million to one hundred billion, revealing the scaling limits of the TCF paradigm. Furthermore, we investigate whether these extremely large LMs can enable a universal item representation for the recommendation task and revolutionize the traditional ID paradigm, which is considered a significant obstacle to developing transferable "one model fits all" recommender models. Our study not only demonstrates positive results but also uncovers unexpected negative outcomes, illuminating the current state of the TCF paradigm within the community. These findings will evoke deep reflection and inspire further research on text-based recommender systems. Our code and datasets will be provided upon acceptance.

## 1 Introduction

The explosive growth of online text data has emphasized the significance of text content recommendation across various domains, including e-commerce, news recommendation, and social media. Text-based collaborative filtering (TCF) has emerged as a pivotal technology for delivering personalized recommendations to users based on textual data, such as product descriptions, reviews, or news articles (Wu et al., 2021; Yuan et al., 2023). The objective of TCF is to accurately capture user preferences and interests from textual data and provide customized recommendations that align with their needs. TCF typically employs language models (LMs) as text encoders, integrated into a recommender architecture using collaborative filtering techniques (Rendle et al., 2010; He et al., 2017; Koren et al., 2009) to generate user-item matching scores (see Figure 1). The promising results of TCF have established it as the mainstream approach for text-based recommendation.

By employing language models (LMs) as item encoders, TCF naturally benefits from the latest advancements in natural language processing (NLP). Particularly, in recent years, large LMs (LLMs) such as GPT-3 (Brown et al., 2020) and ChatGPT (Aiyappa et al., 2023) have achieved revolutionary successes in modeling textual data. However, the text encoders utilized in current TCF models often consist of small or medium-sized LMs, such as word2vec (Mikolov et al., 2013), BERT (Devlin et al., 2018), and RoBERTa (Liu et al., 2019). This limitation may restrict their recommendation capabilities, leading to essential questions: Can TCF achieve exceptional results by leveraging extremely large LMs with tens or hundreds of billions of parameters as text encoders? Is there an upper limit to TCF's performance when pushing the size of the text encoder to its extreme? Can TCF with the LLMs revolutionize the prevailing ID paradigm and usher in a transformative era akin to the universal foundation models (Bommasani et al., 2021) in NLP?

Undoubtedly, the above questions play a crucial role in guiding research within the mainstream TCF paradigm. However, despite numerous TCF algorithms proposed in literature (Wu et al., 2021; Zhang et al., 2021a; Li et al., 2022; Bi et al., 2022; Xiao et al., 2022), none of them have explicitly discussed the above questions. Therefore, instead of introducing yet another algorithm, we aim to decipher the classic TCF models via a series of *audacious experiments* that require immense computational resources.[1] Specifically, we explore the below novel questions.

**Q1: How does the recommender system's performance respond to the continuous increase in the item encoder's size? Is the performance limits attainable at the scale of hundreds of billions?** To answer it, we perform an empirical study where we systematically increase the size of the text encoder from 100 million (100M for short) to 175 billion (175B). This study is conducted on three recommendation datasets, utilizing two most representative recommendation architectures: the two-tower top-N model DSSM (Huang et al., 2013) and a state-of-the-art sequential model SASRec (Kang & McAuley, 2018) with Transformer (Vaswani et al., 2017) as the backbone.

*Novelty clarification*: While the scaling effect has been established in the NLP field, it is important to note that recommender models not only involve the item encoder but also the user encoder. As a result, the potential improvement solely from scaling the item encoder remains unknown. A concurrent[2] preprint (Kang et al., 2023) by Google teams investigated the impact of scaling the item encoder on explicit rating prediction. However, to our best knowledge, we are the first to explore the scaling effect in the context of item recommendation (Cremonesi et al., 2010) from implicit feedback.

**Q2: Can LLMs, such as GPT-3 with 175B parameters, generate universal item representations for recommendation?** Developing universal foundation models is an ambitious goal of NLP, as previos studies have showed the generality of the representations learned by LLMs across various NLP tasks. However, recommender systems (RS) differ from these objective NLP tasks as they are personalized and subjective. This raises the question whether the LLMs pre-trained on non-recommendation data can produce a universal item representation in the recommendation context.

**Q3: Can recommender models with a 175B parameter LLM as the item encoder easily beat the simplest ID embedding based models (IDCF), especially for warm item recommendation?** IDCF is a prevailing recommendation paradigm that has dominated the recommender system (RS) community for over a decade, particularly in the non-cold start setting. It produces high-quality recommendations without relying on any item content information. However, recent studies (Ding et al., 2021; Hou et al., 2022a; Wang et al., 2022; Yuan et al., 2023) indicate that ID features are the key barrier to achieving transferable "one model fits all" recommender models (see Figure 4). This is because IDs, e.g., userID and itemID, are typically not shareable across different practical platforms.

*Novelty clarification*: Although numerous papers claimed that their proposed TCF had achieved state-of-the-art performance, it is recognized that most of these claims are primarily focused on cold-start scenarios (Zhang et al., 2021a). However, in order to truly abandon IDs, it is crucial to surpass its performance in both cold-start and warm scenarios. This presents a considerable challenge because, thus far, no industrial recommender system has dared to claim to completely give up the itemID features (userID can be represented by itemID sequence) in the non-cold start item setting.

**Q4: How close is the TCF paradigm to a universal "one model fits all" recommender model?** In addition to its performance benefits, TCF is often lauded for its potential transferability, allowing for cross-domain and cross-platform recommendations without relying on shared IDs. This advantage contributes to the establishment of a universal foundation model (Bommasani et al., 2021) in the field of recommender systems. Therefore, we aim to study whether TCF, utilizing LLM as the item encoder, exhibits effective transferability, particularly its zero-shot recommendation capability.

If both Q3 and Q4 hold true, LLM will undoubtedly possess the potential to revolutionize the existing recommendation paradigm. In the future, it is conceivable that similar recommendation scenarios could be addressed with a single recommender model, significantly minimizing the need for redundant engineering efforts. However, so far, whether the RS field can develop universal models similar to the NLP community still remains unknown, and the entire community is unable to give a definitive answer. Our primary contribution in this paper is to conduct preliminary research and establish a substantial factual foundation for addressing this question more comprehensively in the near future.

---

[1] Some of our experiments were performed on 32 NVIDIA 80G A100s for several weeks.

[2] When mentioning concurrent work, we specifically mean starting from the earliest available preprints.

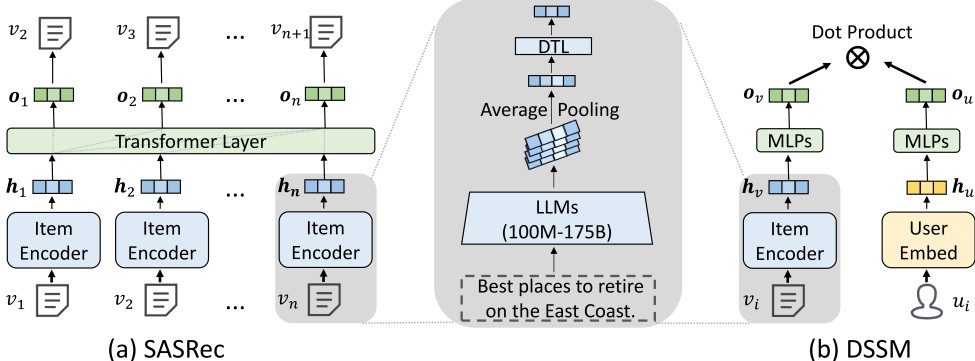

Figure 1: TCF with SASRec and DSSM as recommender backbones. The DTL block is the dense dimension transformation layers. Item or text encoder used in this study can be 175B parameters.

## 2 BACKGROUND

**LMs for Text.** In recent years, significant progress in LM development has had a profound impact on the field of NLP. word2vec, developed in 2013, revolutionized NLP by providing a scalable and efficient way of learning word embeddings. Since then, various improvements have been made to word representation models, such as GloVe (Pennington et al., 2014), TextCNN (Kim, 2015), ELMo (Peters et al., 2018), etc. In 2018, the BERT model showed state-of-the-art performance on a range of NLP tasks by introducing a pre-training approach based on masked language modeling. BERT and its variants (RoBERTa (Liu et al., 2019), ALBERT (Lan et al., 2019), XLNet (Yang et al., 2019), T5 (Raffel et al., 2020), etc.) have become a dominant paradigm in the NLP community in recent years. More recently, ChatGPT, a conversational AI model has gained significant attention for its remarkable performance in a wide range of language tasks. Along this line, several other notable works have contributed to the advancement of LMs, including the Transformer architecture, the GPT (Radford et al., 2018; 2019; Brown et al., 2020) and Llama (Touvron et al., 2023) models.

**LMs for Recommender Systems.** Over the past years, LMs have been widely used in item recommendation tasks, with two main lines of research in this area. The first involves using LMs to represent textual items (Wu et al., 2021; 2019; Zhang et al., 2021a; Yuan et al., 2023; Wu et al.), while the second involves using LMs as user encoders or recommendation backbones, such as SASRec, BERT4Rec (Sun et al., 2019), GRU4Rec (Hidasi et al., 2015), NextItNet (Yuan et al., 2019), and GPT4Rec (Li et al., 2023b). In this paper, we focus primarily on the first line of research. Among the various item encoders, lightweight word2vec and medium-sized BERT are the two most popular options. The literature on this topic can further be classified into two categories: applying pre-extracted textual features (equivalent to a frozen text encoder) (Ding et al., 2021; Bi et al., 2022) and end-to-end (E2E) training of text encoders (Yuan et al., 2023; Yang et al., 2022; Li et al., 2023a). While E2E training typically achieves better results than using a frozen text encoder, the latter approach is much more computationally efficient than E2E training (Yuan et al., 2023).

The success of ChatGPT has prompted the use of prompt techniques for personalized recommendations (Gao et al., 2023; Liu et al., 2023; Dai et al., 2023). This approach can directly utilize the ChatGPT API, eliminating the need for separate model training. It is noteworthy that in recent months, there has been a significant amount of literature on LLM-based recommender systems (see Appendix D), covering a variety of paradigms. However, this paper specifically concentrates on the utilization of LLM as the item encoder.

## 3 PRELIMINARY

We introduce some basic notations and describe two typical recommender paradigms: IDCF & TCF.

*Definition.* We define the set of users as $U = \{u_1, u_2, ..., u_m\}$ and the set of items as $V = \{v_1, v_2, ..., v_n\}$. The user-item interactions are represented by a binary matrix $R = \{r_{uv}\}$, where $r_{uv} \in \{0, 1\}$ indicates whether user $u$ has interacted with item $v$.

Table 1: Dataset characteristics. Bili8M is mainly used for pre-training to answer Q4.

| Dataset | #User | #Item | #Interaction | Item Example |
|---|---|---|---|---|
| MIND | 200,000 | 54,246 | 2,920,730 | Cincinnati Football History (News Title) |
| HM | 200,000 | 85,019 | 3,160,543 | Solid. White. Ladieswear. (Product Description) |
| Bili | 50,000 | 22,377 | 723,071 | The last words of The Humans (Video Title) |
| Bili8M | 8,880,000 | 408,000 | 104,450,865 | The owl is wearing a skirt (Video Title) |

In the standard collaborative filtering (CF) setup, we represent each user by a vector $\theta_u \in \mathbb{R}^k$ and each item by a vector $\beta_v \in \mathbb{R}^k$. The predicted interaction score between user $u$ and item $v$ is computed as $\hat{r}_{uv} = \theta_u^T \beta_v$. To obtain the user and item vectors, we typically optimize a loss function $l(r_{uv}, \theta_u^T \beta_v)$, where $l$ can either be a pairwise BPR (Rendle et al., 2012) loss or a cross-entropy loss.

In the popular ID-based CF (IDCF) models, $\theta_u$ and $\beta_v$, also known as userID and itemID embeddings, can be learned by backpropagating from the user-item interaction data. Following this path, various recommender models have been developed. For instance, if we use a deep neural network to output the user vector $\theta_u$ and the item vector $\beta_v$, denoted by $g(u_i)$ and $h(v_i)$ respectively, the scoring function becomes $\hat{r}_{uv} = g(u_i) \cdot h(v_i)$, which is known as the two-tower DSSM model. Alternatively, if we represent a user by a sequence of $k$ items that she has interacted with, the scoring function is $\hat{r}_{uv} = G(v_1, v_2, ..., v_k)^T \beta_v$, where $G(\cdot)$ is a sequential network, such as SASRec and BERT4Rec.

By utilizing a text encoder $f(v_i)$ to output item representation vectors from the description text, instead of relying on itemID embedding features, the IDCF model can be converted into the TCF model, as depicted in Figure 1. Clearly, the only difference between TCF and the typical IDCF model is in the item representation part. In contrast to IDCF, TCF has the advantage of being able to utilize both item textual content features and user-item interaction feedback data. In theory, the text encoder $f(v_i)$ can take the form of any language model, such as a shallow-layer word2vec model, a medium-sized BERT model, or a super-large GPT-3 model. The text encoder $f(v_i)$ can be either frozen or trained jointly with the whole recommender model in an end-to-end (E2E) fashion.

However, due to computational constraints, most real-world recommender systems adopt a two-stage approach. In this approach, offline features are extracted in advance from a frozen LM encoder and then incorporated as fixed features into the recommender model during both training and inference stages. This is primarily due to the resource-intensive nature of joint or E2E training of text encoders, which requires substantial computing power and time.

## 4 Experimental Setups

### 4.1 Datasets, Models and Evaluation

**Datasets.** We evaluate TCF with LLM as item encoders on three real-world text datasets: the MIND news clicking dataset (Wu et al.), the HM clothing purchase dataset[3], and the Bili[4] comment dataset from an online video recommendation platform. For MIND, we represent items using their news article titles, whereas for HM and Bili, we utilize the respective title descriptions of clothes or videos to represent the items. Across all datasets, each positive user-item interaction is either a click, purchase, or comment, which serves as an implicit indicator of user preference.

Due to memory issues when comparing to E2E training, we constructed interaction sequences for each user by selecting their latest 23 items. We exclude users with fewer than 5 interactions as we do not consider cold user settings. Following the basic pre-processing steps, we randomly selected 200,000 users (along with their interactions) from both the MIND and HM datasets, as well as 50,000

---

[3]https://www.kaggle.com/competitions/h-and-m-personalized-fashionrecommendations/overview

[4]URL: https://www.bilibili.com/. To create this dataset, we randomly crawled short video URLs (with durations of less than 10 minutes) from 23 vertical channels (including technology, cartoons, games, movies, food, fashion, etc.) in Bili. We then extracted the public comments on these videos as positive interactions. Finally, we chronologically combined all user interactions and removed duplicate interactions as the final dataset.

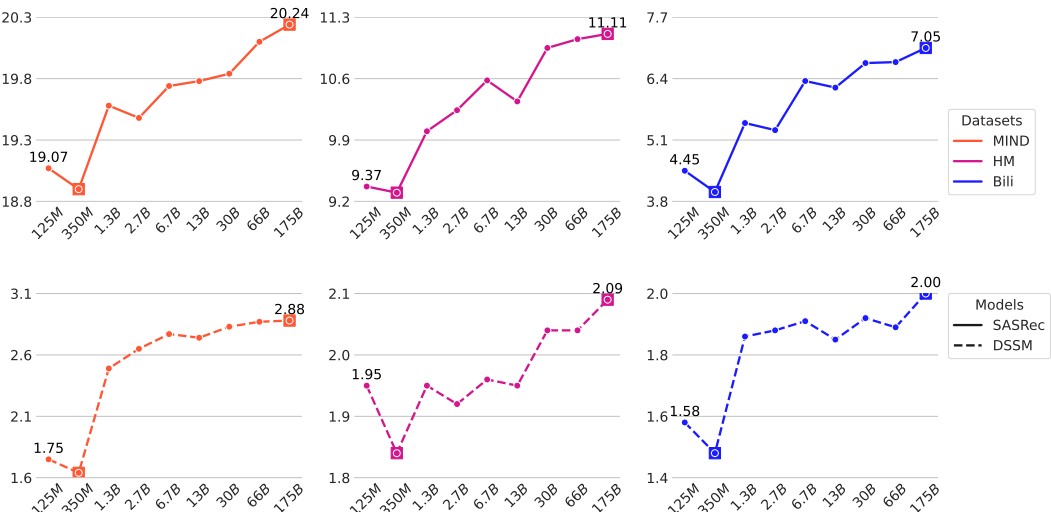

Figure 2: TCF's performance (y-axis: HR@10(%)) with 9 text encoders of increasing size (x-axis). SASRec (upper three subfigures) and DSSM (bottom three subfigures) are used as the backbone.

users from Bili for our main experiments. Additionally, we have also built a large-scale Bili8M (covering the entire Bili dataset) dataset for pre-training purposes to answer Q4.

**Models and Training.** To support our main arguments, we selected two representative recommendation architectures for evaluation: the classical two-tower DSSM model and the SASRec session-based recommender model. Note that as an initial exploration or a limitation of this work, we do not study other more complex models. Despite that, many recommender models can be categorized under the DSSM and SASRec frameworks. For instance, numerous complex CTR (click-through rate) models, despite being single-tower models, are expected to yield similar conclusions as DSSM.[5] .

During training, we utilize the popular batch softmax loss (Yi et al., 2019), which is widely adopted in industrial systems. For text encoders, we evaluated nine different sizes of GPT models, ranging from 125M to 175B parameters. These GPT models were re-implemented by Meta AI and are interchangeably referred to as OPT (Zhang et al., 2022). As for hyper-parameters, we first perform a grid search for standard IDCF as a reference, After determining the optimal hyper-parameters for IDCF, we search them for TCF around these optimal values. We report details in Appendix B.

**Evaluation.** We evaluate the performance of all models using two popular top-K ranking metrics, namely HR@10 (Hit Ratio) and NDCG@10 (Normalized Discounted Cumulative Gain) (Yuan et al., 2023). NDCG@10 is reported in Appendix C for saving space. The latest user-item interaction was used for evaluation, while the second-to-last interaction was used for hyper-parameter searching, and all other interactions were used for training. All items in the pool are used for evaluation, suggested by (Krichene & Rendle, 2020).

## 5 Q1: HAS THE TCF PARADIGM HIT A PERFORMANCE CEILING?

To answer Q1, we conduct experiments by increasing the size of text encoders in the TCF models, ranging from 125M to 175B parameters. We use SASRec and DSSM as recommender backbones. The results are given in Figure 2. All LMs are frozen for this study.

As shown, TCF models generally improve their performance by increasing the size of their text encoders. For instance, with the SASRec as the backbone, TCF improved the recommendation accuracy from 19.07 to 20.24 on MIND, from 9.37 to 11.11 on HM, and from 4.45 to 7.05 on Bili,[6] resulting in improvements of 6.1%, 18.6%, and 58.4%, respectively. Similar observations can also be made for the DSSM backbone. Furthermore, based on the observed performance trend, we can

---

[5]This difference between single or two towers does not affect our key findings.

[6]We also conduct this study on the large-scale Bili8M dataset, as shown in Appendix Figure 8.

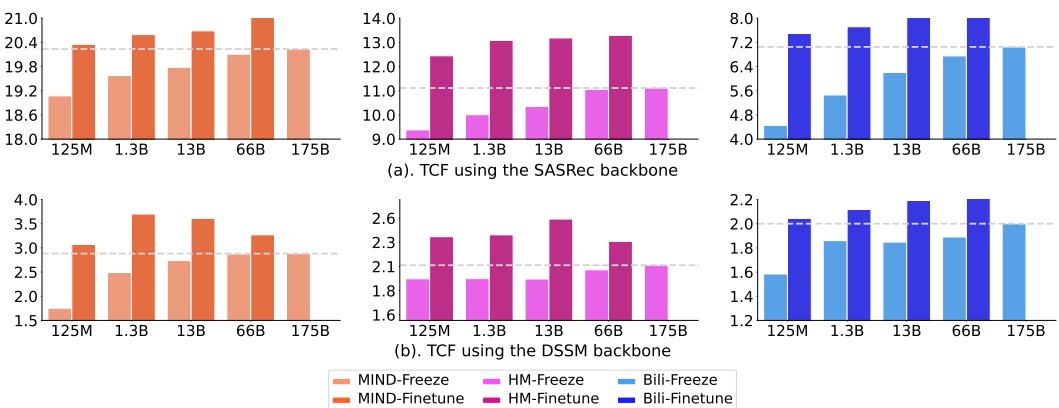

Figure 3: TCF with retrained LM vs frozen LM (y-axis: HR@10(%)), where only the top two layers are retrained. The 175B LM is not retrained due to its ultra-high computational cost.

conclude that the TCF models' performance has not yet converged when increasing the size of their text encoders, such as from 13B to 175B. These results suggest that **(answer to Q1) the TCF model with a 175B parameter LM may not have reached its performance ceiling**. In other words, if we had an even larger LM as the text encoder, TCF's performance could potentially be further improved. This is a highly desirable property because it indicates that **using more powerful LMs (if developed in the future) as text encoders can result in higher recommendation accuracy.**

Interestingly, we find that the TCF model with the 350M parameter LM exhibits the poorest performance across all three datasets, regardless of whether it uses the DSSM or SASRec backbone. However, the 350M LM is not the smallest text encoder. This could happen because the scaling relationship between text encoder size and performance is not necessarily strictly linear. However, by examining the pre-training code and official documentation, we discover that the 350M-parameter OPT was implemented with several differences compared to all other versions.[7] This provides an explanation for our results. Additionally, beyond the discussion scope of this paper, we also note that TCF utilizing the SASRec backbone shows significantly superior performance compared to TCF with the DSSM backbone. Similar findings were reported in much previous literature (Yuan et al., 2023; Sun et al., 2019; Zhou et al., 2020). One possible reason for this is that representing users using their interacted items is more effective than using solely the userID feature. Another reason could be that the SASRec architecture, based on the sequence-to-sequence (seq2seq) training approach, is more powerful than the two-tower DSSM architecture.

## 6 Q2: CAN THE 175B LLM ACHIEVE UNIVERSAL TEXT REPRESENTATION?

We are curious about whether a LM with 175B parameters possess a degree of universality in text encoding. Unlike the objective NLP tasks, here we examine this property using personalized recommendation as a downstream task. Assuming that a $k$-dimensional text representation $\beta_v$ encoded by the 175B parameter LM is an ideal universal representation, any application involving text representation can directly choose a subset or the entire set of features from $\beta_v$ by providing a weight vector $w$ that represents the importance of these elements, i.e., $y = w^T \beta_v$. For example, in a basic matrix factorization setting, $w$ could represent user preference weight to item features, i.e. $w = \theta_u$. If all factors of user preference can be observed by the features in $\beta_v$, we only need to learn their linear relationship. Moreover, for a perfect universal vector $\beta_v$, using a frozen representation should be just as effective as fine-tuning it on a new dataset, or even superior to fine-tuning.

Based on the analysis, we can simply compare the frozen item representation with the fine-tuned item representation to verify our question. Note that previous studies such as (Yuan et al., 2023) have investigated this issue, but they only examined text encoders with a size of 100M parameters. Given

---

[7]For instance, in all other pre-trained models, the layernorm layer is implemented before the attention layer, while in the 350M model, it is opposite. Plus, its embedding & hidden layer dimensions are also set differently.

Table 2: Accuracy comparison (HR@10) of IDCF and TCF using the DSSM & SASRec backbones. *FR* is TCF using frozen LM, while *FT* is TCF using fine-tuned LM.

| Data | SASRec | | | DSSM | | |
|------|--------|--------|--------|--------|--------|--------|
| | IDCF | $175B^{FR}$ | $66B^{FT}$ | IDCF | $175B^{FR}$ | $66B^{FT}$ |
| MIND | 20.05 | 20.24 | **21.07** | **3.99** | 2.83 | 3.27 |
| HM | 12.02 | 11.24 | **13.29** | **6.79** | 2.09 | 2.35 |
| Bili | 7.01 | 6.88 | **8.15** | **2.27** | 2.00 | 2.01 |

Table 3: Zero-shot recommendation accuracy (HR@10). $175B_{zero}$ means zero-shot accuracy of TCF with 175B LM. 'train' is to retrain TCF on these data.

| Model | MIND | HM | QB |
|-------|------|------|------|
| Random | 0.02 | 0.01 | 0.18 |
| $175B_{zero}$ | 0.13 | 0.39 | 4.30 |
| $175B_{train}$ | 20.24 | 11.11 | 29.90 |

Table 4: Warm item recommendation (HR@10). 20 means items < 20 interactions are removed. $TCF_{175B}$ uses the pre-extracted features from the 175B LM. Only the SASRec backbone is reported.

| Data | MIND | | | HM | | | Bili | | |
|------|------|------|------|------|------|------|------|------|------|
| #Interaction | 20 | 50 | 200 | 20 | 50 | 200 | 20 | 50 | 200 |
| IDCF | 20.56 | 20.87 | 23.04 | 13.02 | 14.38 | 18.07 | 7.89 | 9.03 | 15.58 |
| $TCF_{175B}$ | 20.59 | 21.20 | 22.85 | 12.03 | 12.68 | 16.06 | 7.76 | 8.96 | 15.47 |

the significantly enhanced representation capabilities of the 175B LM (as shown in Table 5), it is uncertain whether the findings remain consistent when the encoder is scaled up by a factor of 1000

As shown in Figure 3, TCF models (both SASRec and DSSM) outperform their frozen versions when the text encoders are retrained on the recommendation dataset. Surprisingly, TCF with a fine-tuned 125M LM is even more powerful than the same model with a frozen 175B LM. This result potentially suggests that **(answer to Q2) even the item representation learned by an extremely large LM (e.g., GPT-3) may not result in a universal representation, at least not for the text recommendation task.** Another key insight is that although LLMs have revolutionized so many NLP problems, there is still a significant domain gap between RS and NLP - specifically, inferring user preferences appears to be more challenging. We suspect that the text representation even extracted from the strongest and largest LM developed in the future may not perfectly adapt to the RS dataset. Retraining the LLM on the target recommendation data is necessary for optimal results. However, from a positive perspective, since LLMs have not yet reached the performance limit, if future more powerful LLMs are developed, the performance of frozen text representation may become more close to fine-tuning. For instance, we observe that SASRec with a 175B LM (compared to the 125M LM) is already very close in performance to the fine-tuned 66B LM, with relative accuracy gaps of 3.92%, 16%, 13.5% on HM, and Bili, respectively. This is a promising discovery since fine-tuning such a large LM is very challenging in practical scenarios.[8] Note while we did not fine-tune all layers of the largest LM, we did assess the performance using medium-sized LMs (such as 1.3B and 13B) by optimizing all layers and the top two layers, which yielded comparable results.

It is worth noting that the above conclusions are based on the assumption that user-item interaction feedback serves as the gold standard for the recommendation task, but this may not always be the case in practice. As a limitation, this study does not address this issue, as the entire theory of modern recommender systems is currently based on this assumption.

## 7 Q3: CAN IDCF BE EASILY SURPASSED BY TCF WITH A 175B LLM?

TCF is a classical paradigm for text-based recommendation, while IDCF is the dominant paradigm in the entire field of RS. Can TCF models with a 175B parameter LLM easily beat IDCF models with learnable item embedding vectors? While many prior studies have reported that their TCF models achieved state-of-the-art results, few have explicitly and fairly compared their models with

---

[8] Even when fine-tuning only the top two layers, as used in our experiments, it still necessitates 10-100x more training time than using pre-extracted fixed features.

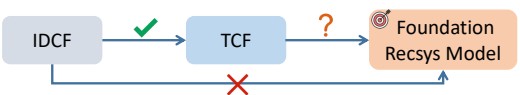

Figure 4: Route to foundation recommender models (FRM). The cross indicates that the IDCF paradigm have no chance to achieve FRM, the tick indicates that for text-centric RS, TCF can basically replace IDCF, and the question mark indicates that whether the TCF paradigm can achieve the widely recognized FRM remains still unknown.

Table 5: TCF's results (HR@10) with representative text encoders in the last 10 years. Text encoders are frozen and the SASRec backbone is used. Advances in NLP benefit RS.

| Model | Date | MIND | HM | Bili |
|---|---|---|---|---|
| word2vec | 2013 | 15.21 | 8.08 | 2.66 |
| $BERT_{large}$ | 2018 | 18.99 | 9.68 | 3.56 |
| $T5_{XXL}$ | 2019 | 19.56 | 9.21 | 4.81 |
| $OPT_{175B}$ | 2022 | 20.24 | 11.11 | 7.05 |

corresponding IDCF counterparts *under the same backbone networks and experimental settings (including samplers and loss functions).*[9] Moreover, many of them focus on cold item setting, with fewer studies explicitly examining regular (with both cold and warm items) or warm item settings. Recently, (Yuan et al., 2023) discovered that TCF can be comparable to IDCF by jointly training a 100M parameter LM, but frozen representations still significantly underperformed. Therefore, a natural question is whether our conclusions would differ if we use a 1000x larger LLM as the item encoder?

As shown in Table 2, we observe that even with the 175B parameter LLM and fine-tuned 66B parameter LLM, TCF is still substantially inferior to IDCF when using DSSM as the backbone. These results are consistent with (Yuan et al., 2023). As explained, the DSSM architecture and training approach exhibit limited effectiveness in training TCF models. Both the IDCF and TCF models with DSSM perform worse than the seq2seq-based SASRec model. However, a notable finding different from (Yuan et al., 2023) is that we reveal that TCF with the SASRec backbone performs comparably to IDCF on the MIND and Bili datasets, even when the LLM encoder is frozen, as shown in Table 2 and 4. This represents a significant advancement since no previous study has *explicitly* claimed that TCF, by freezing an NLP encoder (or utilizing pre-extracted fixed representations), can achieve on par performance to its IDCF counterparts specifically in the context of warm item recommendation.[10] This is probably because smaller LM-based item encoders in prior literature, such as BERT and word2vec, are inadequate in generating effective text representations comparable to IDCF, see Table 5.

The reason for the weaker performance of TCF on HM is that textual information alone is insufficient to fully represent the product item, as factors such as price and quality are also critical in enticing user clicks and purchases on HM. However, in the case of news recommendation, we can generally assume that users are primarily drawn to the textual content (i.e., titles) of items, although this may not always be the case. That is the reason we believe TCF with frozen text encoders performs on par with IDCF is surprising as IDCF can implicitly learn latent factors beyond textual features but feature representation pre-extracted from a NLP encoder cannot. Furthermore, we notice that SASRec with a fine-tuned text encoder can clearly outperform IDCF on all three datasets. However, as mentioned, such end-to-end training using a text encoder is computationally expensive, despite its effectiveness.

**The answer to Q3 is that, for *text-centric* recommendation, TCF with the seq2seq based SASRec backbone and utilizing a 175B parameter frozen LLM can achieve similar performance to standard IDCF, even for warm item recommendation. However, even by retraining a super-large LM item encoder, TCF with a DSSM[11] backbone has little chance to compete with its corresponding IDCF. The simple IDCF still remains a highly competitive approach in the warm item recommendation setting.** If the computation can be reduced, joint training of a powerful sequential recommender model (i.e., SASRec) with its text encoder can lead to markedly better results than IDCF.

---

[9]Without conducting a fair comparison, researchers are unable to accurately assess the true progress.

[10]We simply omit the results for cold item recommendation, as TCF has been consistently demonstrated to outperform IDCF in these settings in numerous literature, e.g., in (Yuan et al., 2023; Hou et al., 2022b).

[11]A very recent study (Rajput et al., 2023) suggested that standard CTR models, such as DSSM and DeepFM (Guo et al., 2017), may be replaced by the seq2seq generative architecture, such as SASRec. This means seq2seq model may have a chance to be a mainstream recommendation architecture.

# 8 Q4: HOW CLOSE IS THE TCF PARADIGM TO A UNIVERSAL RECOMMENDER MODEL?

In this paper, we are particularly interested in comparing with the dominant IDCF paradigm. This is because ID features (including userIDs and itemIDs) are considered as a primary obstacle to the transferable or foundation recommender models due to their non-sharability (Yuan et al., 2023; Hou et al., 2022a; Rajput et al., 2023; Wang et al., 2022; Ding et al., 2021; Shin et al., 2021). We argue that to achieve foundation models in recommender systems may require satisfying two conditions, as illustrated in Figure 4: (1) abandoning userID and itemID features, and (2) achieving effective transferability across domains and platforms. Based on the above results, we conclude that for text-centric recommender systems, TCF-based sequential recommender models can basically substitute IDCF methods. However, regarding (2), it remains uncertain whether TCF has impressive transfer learning ability, especially when its item representations are extracted from a extremely large LM.

Inspired by the remarkable success of zero-shot learning in NLP, our goal is to assess the zero-shot transfer learning capability of TCF, considering that items with text features may be inherently transferable. Following (Ding et al., 2021), we first pre-train a SASRec-based TCF model with the 175B parameter frozen LM as item encoder on the large-scale Bili8M dataset. We then directly evaluate the pre-trained model in the testing set of MIND, HM and QB[12]. The results, presented in Table 3, indicate that while TCF models outperform random item recommendation by achieving an accuracy improvement of 6-40x, they still fall notably short of TCF models that have been retrained on the new data. We note that user behaviors in the source Bili8M dataset may differ significantly from HM and MIND datasets due to their distinct contexts of e-commerce and news recommendation scenarios. However, it is similar to that of QB, as both involve similar types of item recommendations.

**The answer to Q4 is that while TCF models with LLMs do exhibit a certain degree of transfer learning capability, they still fall significantly short of being a universal recommender model, as we had initially envisioned.** For a universal recommendaton model, not only should item representations be transferable, but also the matching relationship between users and items needs to be transferable. However, the matching relationship is closely related to the exposure strategy or bias of the specific recommender platform. Therefore, compared to NLP and computer vision (CV), the transferability of recommender models is even more challenging. This also explains why, up until now, there haven't been any pre-trained models in the field of recommender systems that have attained the same level of prominence and recognition as BERT and ChatGPT in the NLP field. For instance, the lack of a pre-trained recommender model in the HuggingFace library that can support various recommendation scenarios (similar or dissimilar) further reinforces this point. However, this does not necessarily indicate that TCF have no potential to become a universal recommender model. It will require the collective effort of the entire recommender system community. This includes utilizing highly diverse and extremely large pre-training datasets (Ni et al., 2023), employing advanced training and transfer learning techniques, and engaging in deeper considerations for evaluation (e.g., removing platform exposure bias when evaluating downstream recommendation tasks (Fan et al., 2023)).

# 9 CONCLUSION

This paper does not describe a new text recommender algorithm. Instead, it extensively explores the performance limits and several core issues of the prevailing text-based collaborative filtering (TCF) techniques. From a positive perspective, TCF still has untapped potential and can further improve with the advancement of NLP large models' representation capacity. However, on the other hand, even with item encoders consisting of tens of billions of parameters, re-adaptation to new data remains necessary for optimal recommendations. Furthermore, the current state-of-the-art TCF models do not exhibit the anticipated strong transferability, suggesting that building large foundation recommender models may be more challenging than in the fields of NLP and computer vision. Nonetheless, TCF with text encoders of 175 billion parameters is already a significant leap forward, as it fundamentally challenges the dominant ID-based CF paradigm, which is considered the biggest obstacle to developing universal "one-for-all" recommender models, although not the only one.

---

[12]QQ Browser (QB) is a feed recommendation dataset from which we extracted short video titles, similar to items from Bili. It contains 5546 items 17809 users and 137979 interactions.

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

## A  TEXT ENCODER DETAILS

Table 6: List of Large LMs and their details

| Name | Model Size | Parameters | Architecture | Source |
|---|---|---|---|---|
| BERT | Large | 340M | Encoder-only | `https://huggingface.co/bert-large-uncased` |
| T5Encoder | XXL | 5.5B | Encoder-only | `https://huggingface.co/t5-11B` |
| OPT | 125M | 125M | Decoder-only | `https://huggingface.co/facebook/opt-125m` |
| | 350M | 350M | Decoder-only | `https://huggingface.co/facebook/opt-350m` |
| | 1.3B | 1.3B | Decoder-only | `https://huggingface.co/facebook/opt-1.3b` |
| | 2.7B | 2.7B | Decoder-only | `https://huggingface.co/facebook/opt-2.7b` |
| | 6.7B | 6.7B | Decoder-only | `https://huggingface.co/facebook/opt-6.7b` |
| | 13B | 13B | Decoder-only | `https://huggingface.co/facebook/opt-13b` |
| | 30B | 30B | Decoder-only | `https://huggingface.co/facebook/opt-30b` |
| | 66B | 66B | Decoder-only | `https://huggingface.co/facebook/opt-66b` |
| | 175B | 175B | Decoder-only | `https://github.com/facebookresearch/metaseq/tree/main/projects/OPT` |

## B  HYPER-PARAMETER TUNING

Before tuning hyper-parameters for TCF, we grid search IDCF on each dataset as a reference. Specifically, we search for learning rates within the range of {*1e-3, 1e-4, 1e-5, 5e-5*} and hidden dimensions from {*64, 128, 256, 512, 1024*} for both DSSM and SASRec; we search batch size within {*64, 128, 256, 512*} for SASRec and {*1024, 2048, 4096*} for DSSM; we set a fixed dropout rate of 0.1, and tune the weight decay within {*0.01, 0.1*}; we search the number of Transformer layers in SASRec within {*1, 2, 3, 4*}, and the number of attention heads within {*2, 4, 8*}. After determining the optimal hyper-parameters for IDCF, we search the TCF around these optimal values with the frozen text encoder (using the 125M variant) by the same stride. To ensure a fair comparison of the scaling effect, we employ the same hyper-parameters for all TCF models with different sizes of frozen text encoder (i.e., pre-extracted features). For TCF models with E2E learning of text encoders, we kept the optimal hyper-parameters the same as those with frozen encoder, except for the learning rates. We separately tune the learning rate, as larger text encoders typically require a smaller learning rate. The details are given below. We utilize the AdamW optimizer (Loshchilov & Hutter, 2017) for all models.

Table 7: Optimal hyper-parameters for IDCF, including learning rate ($lr$), embedding size ($k$), batch size ($bs$), the number of Transformer layers ($l$), the number of attention heads ($h$), and weight decay ($wd$). The dimension of feed forward layer in Transformer block is $4 \times k$.

| Data | SASRec | | | | | | DSSM | | | | | |
|---|---|---|---|---|---|---|---|---|---|---|---|---|
| | $lr$ | $k$ | $bs$ | $l$ | $h$ | $wd$ | $lr$ | $k$ | $bs$ | $l$ | $h$ | $wd$ |
| MIND | 1e-4 | 512 | 64 | 2 | 2 | 0.1 | 1e-5 | 256 | 4096 | 2 | 2 | 0.1 |
| HM | 1e-3 | 128 | 128 | 2 | 2 | 0.1 | 1e-4 | 1024 | 1024 | 2 | 2 | 0.1 |
| Bili | 1e-3 | 128 | 256 | 2 | 2 | 0.1 | 1e-3 | 1024 | 1024 | 2 | 2 | 0.1 |

Table 8: Optimal hyper-parameters for TCF with frozen text encoder.

| Data | SASRec | | | | | | DSSM | | | | | |
|---|---|---|---|---|---|---|---|---|---|---|---|---|
| | $lr$ | $k$ | $bs$ | $l$ | $h$ | $wd$ | $lr$ | $k$ | bs | $l$ | $h$ | $wd$ |
| MIND | 1e-4 | 512 | 64 | 2 | 2 | 0.1 | 1e-5 | 256 | 4096 | 2 | 2 | 0.1 |
| HM | 1e-4 | 512 | 64 | 2 | 2 | 0.1 | 1e-3 | 1024 | 1024 | 2 | 2 | 0.1 |
| Bili | 1e-3 | 128 | 64 | 2 | 2 | 0.1 | 1e-3 | 512 | 1024 | 2 | 2 | 0.1 |

Table 9: The learning rate of item encoder for TCF with E2E learning. The search range is suggested by the original paper of OPT.

| Data | SASRec | | | | DSSM | | | |
|------|------|------|------|------|------|------|------|------|
| | 125M | 1.3B | 13B | 66B | 125M | 1.3B | 13B | 66B |
| MIND | 1e-4 | 1e-4 | 8e-5 | 3e-5 | 1e-4 | 1e-4 | 1e-4 | 1e-4 |
| HM | 1e-4 | 1e-4 | 1e-4 | 8e-5 | 1e-4 | 1e-4 | 1e-4 | 1e-4 |
| Bili | 1e-4 | 1e-4 | 3e-5 | 3e-5 | 1e-4 | 1e-4 | 1e-4 | 1e-4 |

## C  MORE RESULTS ON NDCG@10

Table 10: Warm item recommendation (NDCG@10). 20 means items < 20 interactions are removed. $TCF_{175B}$ uses the pre-extracted features from the 175B LM. Only SASRec backbone is reported.

| Data | MIND | | | HM | | | Bili | | |
|------|------|------|------|------|------|------|------|------|------|
| #Inter. | 20 | 50 | 200 | 20 | 50 | 200 | 20 | 50 | 200 |
| IDCF | 11.36 | 11.47 | 12.71 | 8.47 | 9.35 | 12.07 | 4.41 | 5.01 | 8.30 |
| $TCF_{175B}$ | 11.38 | 11.61 | 12.56 | 7.44 | 7.90 | 10.33 | 4.34 | 4.84 | 7.97 |

Table 11: Accuracy (NDCG@10) comparison of IDCF and TCF using DSSM and SASRec. *FR* represents using frozen LM, while *FT* represents using fine-tuned LM.

| Data | Metric | SASRec | | | DSSM | | |
|------|--------|--------|--------|--------|--------|--------|--------|
| | | ID | $TCF_{175B}^{FR}$ | $TCF_{66B}^{FT}$ | ID | $TCF_{175B}^{FR}$ | $TCF_{66B}^{FT}$ |
| MIND | NDCG@10 | 11.06 | 11.09 | **11.77** | **1.72** | 1.42 | 1.58 |
| HM | NDCG@10 | 7.76 | 6.91 | **8.20** | 4.19 | 1.08 | 1.22 |
| Bili | NDCG@10 | 3.93 | 3.77 | **4.56** | 1.12 | 1.01 | 1.06 |

Table 12: Zero-shot recommendation accuracy (NDCG@10). $175B_{zero}$ means zero-shot accuracy of TCF with 175B LM. 'train' is to retrain TCF on these data.

| Model | Date | MIND | HM | Bili |
|-------|------|------|------|------|
| Word2vec | 2013 | 7.52 | 4.81 | 1.30 |
| $BERT_{large}$ | 2018 | 10.45 | 6.01 | 1.83 |
| $T5_{XXL}$ | 2019 | 10.72 | 5.50 | 2.54 |
| $OPT_{175B}$ | 2022 | 11.17 | 6.88 | 3.95 |

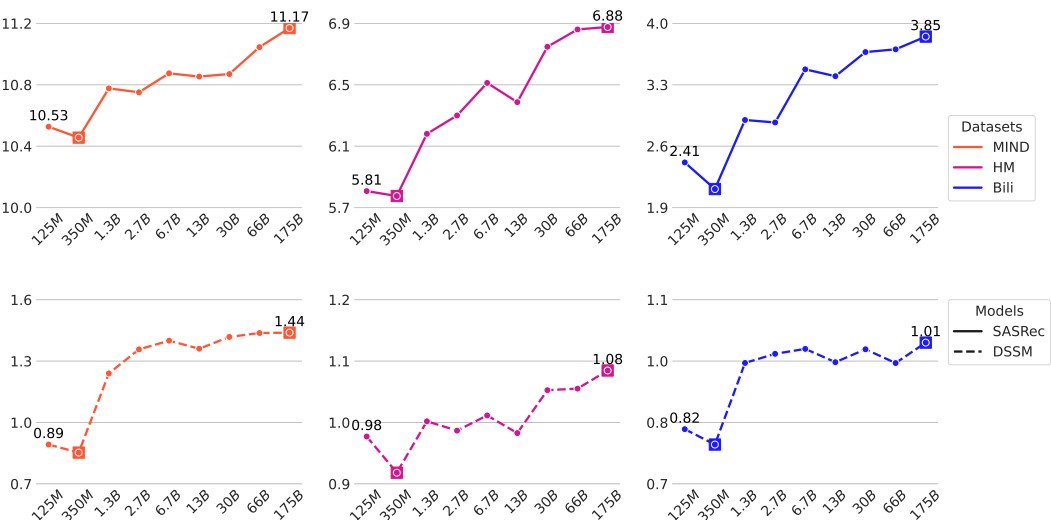

Figure 5: TCF's performance (y-axis: NDCG@10(%)) with 9 text encoders of increasing size (x-axis). SASRec (upper three subfigures) and DSSM (bottom three subfigures) are used as the backbone.

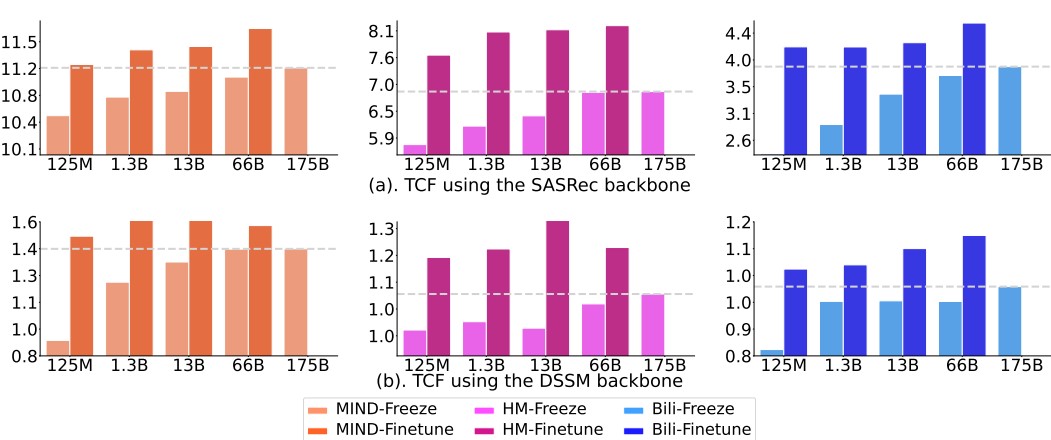

Figure 6: TCF with retrained LM vs frozen LM (y-axis: NDCG@10(%)), where only the top two layers are retrained. The 175B LM is not retrained due to its ultra-high computational cost.

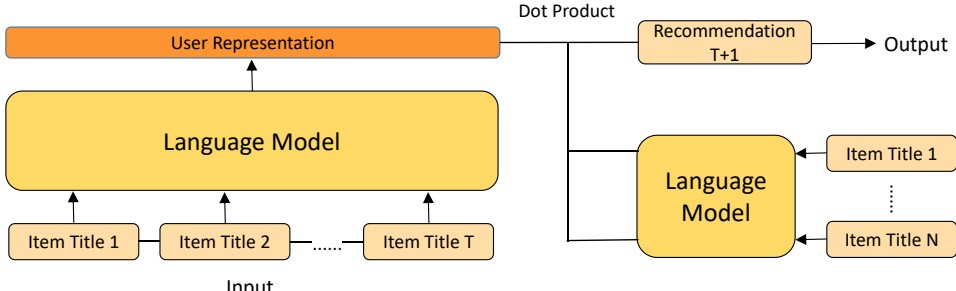

Figure 7: Architecture of GPT4Rec.

## D OTHER PARADIGMS FOR LLM-BASED RECOMMENDER MODELS

This paper primarily focuses on the TCF paradigm with LLMs as item encoders. However, apart from TCF, there are other paradigms for LLM-based recommendation models. Here, we briefly investigate two popular approaches, namely the GPT4Rec paradigm and ChatGPT4Rec paradigm (see next section)

The GPT4Rec (Li et al., 2023b) paradigm (as illustrated in Fig 7) utilizes LLM as the backbone architecture rather than the item encoder. In this approach, the text of items clicked by users is concatenated and fed into the LLM to obtain user representations. Recommendations are then made by calculating the dot product between the user representation and the candidate item representations, which are also represented using LLM.

We conducted experiments using LLMs with 1.3B and 125M versions. As shown in Table 13, fine-tuning only the top-1 block resulted in significantly worse performance compared to full fine-tuning. Even the 1.3B version LLM performed substantially worse than the 125M version when fully fine-tuned. In fact, we have discovered that freezing the LLM or only fine-tuning the top-1 block makes it extremely challenging to provide effective recommendations using this approach.

Furthermore, the GPT4Rec paradigm necessitates significant computational resources and GPU memory. When dealing with longer user sequences, it is not practical to fully fine-tune a very large LLM. This limitation helps explain why the GPT4Rec paradigm has not yet employed very large LLMs as a backbone. Most of such paper used the LLM with a size smaller than 3B.

Furthermore, when comparing the performance of the GPT4Rec model with our TCF approach, it becomes apparent that the new GPT4Rec paradigm significantly underperforms compared to the classical TCF paradigm.

Table 13: Results of GPT4Rec (HR@10(%)) paradigm. Even in 80G A100, we were not able to fully fine-tune 1.3B GPT4Rec. Note that this paradigm requires too much computation and memory when there are long-range item interactions.

| Dataset | Finetune 1 block (125M) | Finetune 1 block (1.3B) | Finetune All (125M) |
|---------|-------------------------|-------------------------|---------------------|
| MIND | 3.26 | 5.19 | 13.48 |
| Bili | 0.04 | 0.09 | 1.28 |
| HM | 0.12 | 0.17 | 2.89 |

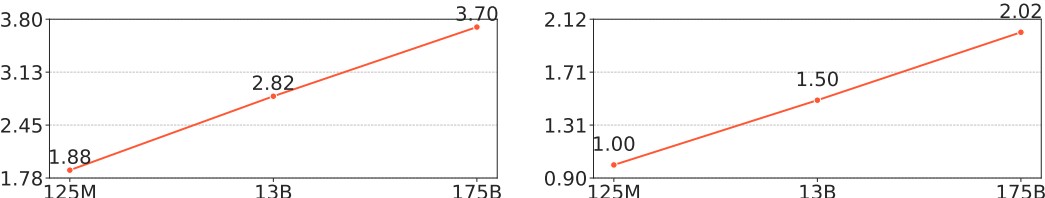

Figure 8: TCF's performance (y-axis: HR@10(%) in left and NDCG@10(%) in right) of 3 item encoder with increased sizes (x-axis) on Bili8M. SASRec is used as the backbone. LLM is frozen.

## E  CHATGPT4REC

Beyond the TCF paradigm, building text recommender models by leveraging prompt strategies is also becoming increasingly popular (Geng et al., 2023; Wang & Lim, 2023; Li et al., 2023c; Zhang et al., 2021b). Recently, due to the tremendous success of ChatGPT, a number of preprint papers have explored the use of prompt engineering with ChatGPT for recommender systems (Gao et al., 2023; Liu et al., 2023; Dai et al., 2023; Wang et al., 2023). Readers may be interested in whether prompt-based techniques on ChatGPT, referred to as ChatGPT4Rec[13], can outperform the classical TCF paradigm under the common recommendation setting. Do we still need the TCF paradigm in the ChatGPT era?

We randomly selected 1024 users from the testing sets of MIND, HM, and Bili, and created two tasks for ChatGPT. In the first task (Task 1 in Table 14), ChatGPT was asked to select the most preferred item from four candidates (one ground truth and three randomly selected items), given the user's historical interactions as a condition. The second task (Task 2 in Table 14) was to ask ChatGPT to rank the top-10 preferred items from 100 candidates (one ground truth and 99 randomly selected items, excluding all historical interactions), also provided with the user's historical interactions as input. We begin by asking ChatGPT if it understands the request, in order to ensure the quality of the prompts. Both the prompts and their answers in Fig 9 to Fig 12. The results are given in Table 14, which illustrate ChatGPT's poor performance compared to TCF in typical recommendation settings. Similar bad results have also been reported in (Liu et al., 2023; Bao et al., 2023). Despite that, we believe with more finely-tuned prompts, ChatGPT may have the potential for certain recommendation scenarios. Another major drawback of ChatGPT is that it cannot generate recommendations from an

---

[13]We use gpt-3.5-turbo API in https://platform.openai.com/docs/models/gpt-4

Table 14: ChatGPT4Rec vs TCF. *FR & FT* means freezing and fine-tuning LM respectively.

| Data | Task 1-HR@1 | | | | Task 2-HR@10 | | | |
|------|--------|---------|-------------------|------------------|--------|---------|-------------------|------------------|
|      | Random | ChatGPT | $\text{TCF}_{175B}{}^{FR}$ | $\text{TCF}_{66B}{}^{FT}$ | Random | ChatGPT | $\text{TCF}_{175B}{}^{FR}$ | $\text{TCF}_{66B}{}^{FT}$ |
| MIND | 25.00  | 25.68   | 96.48             | 96.58            | 10.00  | 9.86    | 97.07             | 97.9             |
| HM   | 25.00  | 29.59   | 88.18             | 90.63            | 10.00  | 12.21   | 83.79             | 90.33            |
| Bili | 25.00  | 24.51   | 77.64             | 81.05            | 10.00  | 8.50    | 70.80             | 73.34            |

item pool with millions of items due to limited memory. This limitation limits the use of ChatGPT as a re-ranking module in existing recommendation pipelines and prevents its use for recommendations from a huge pool of millions of closed-domain items.

In recent months, there has been a substantial increase in literature on recommender systems based on LLM and ChatGPT. It is challenging to thoroughly investigate and compare all these approaches within a single paper. However, we believe that these new paradigms (fine-tuning, prompt-tuning, instruct-tuning, adapter-tuning, etc.) have the potential to bring fresh insights to the recommender system community and may even surpass existing classical paradigms.

The primary contribution of this paper focuses on the performance of the TCF paradigm, which is defined to employ the LLM as the item encoder.

## F  THE PROMPTS OF CHATGPT4REC

The output by ChatGPT in Figure 9 indicates that ChatGPT fully understands the recommendation request. Figure 10, 11 and 12 are prompts for ChatGPT on MIND, HM, and Bili respectively.

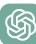

```
Do you understand our request, is it clear? "Please return a list of the top 10 most relevant
video titles from the unordered candidate pool as a personalized recommendation, based on the
user's viewed history, formatted as a JSON array. Do not include any titles from the user's
viewed history in the recommendations. Please note that the candidate pool is not in any
particular order, any ranking based on the order of items in the pool should be assumed.

User viewed history:
1.MC Lucky Gun PVP! When you fire a gun in "Desert Ash" and duel with lucky cubes!
2.Minecraft Kep1er WADADA Dance Relay Open
...
12.Central Conservatory of Music professors perform famous songs! The roof beautifully.

Unordered candidate news title pool (no order should be assumed):
- Times Youth League Ma Jiaqi
- Teach you to make homemade strawberry ice cream,with delicate taste and no ice residue,
...
- Pea Shooter: There's something wrong with the zombies here today!

Top 10 news titles from candidates pool to recommend to the user as JSON array (from highest
to lowest priority, titles only): ".
```

```
Yes, I understand your request. You would like to receive a list of the top 10 most relevant
video titles as personalized recommendations for a user. The recommendations should be based
on the user's viewed history, and the titles should not include any videos that the user has
already viewed. The candidate pool is an unordered list, and any ranking based on the order
of items in the pool should be assumed.To clarify, you have provided a sample user viewed
history and an unordered candidate video pool. You are requesting the top 10 video titles
from the candidate pool to be recommended to the user. The recommendations should be returned
as a JSON array, sorted from highest to lowest priority based on relevance. Only the titles
should be included in the JSON array.Is that correct?
```

Figure 9: Verifying that ChatGPT understands the request.

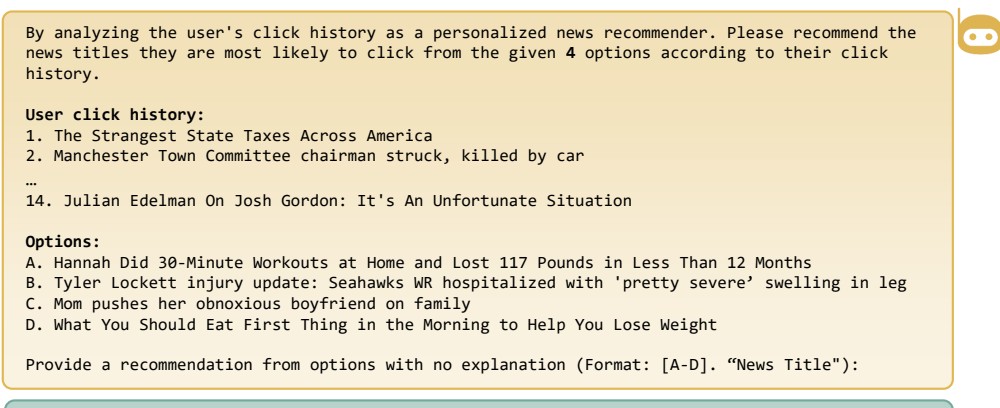

By analyzing the user's click history as a personalized news recommender. Please recommend the news titles they are most likely to click from the given **4** options according to their click history.

**User click history:**
1. The Strangest State Taxes Across America
2. Manchester Town Committee chairman struck, killed by car

…
14. Julian Edelman On Josh Gordon: It's An Unfortunate Situation

**Options:**
A. Hannah Did 30-Minute Workouts at Home and Lost 117 Pounds in Less Than 12 Months
B. Tyler Lockett injury update: Seahawks WR hospitalized with 'pretty severe' swelling in leg
C. Mom pushes her obnoxious boyfriend on family
D. What You Should Eat First Thing in the Morning to Help You Lose Weight

Provide a recommendation from options with no explanation (Format: [A-D]. "News Title"):

B. Tyler Lockett injury update: Seahawks WR hospitalized with 'pretty severe' swelling in leg

(a) Example of Task 1 in Table 14) for MIND

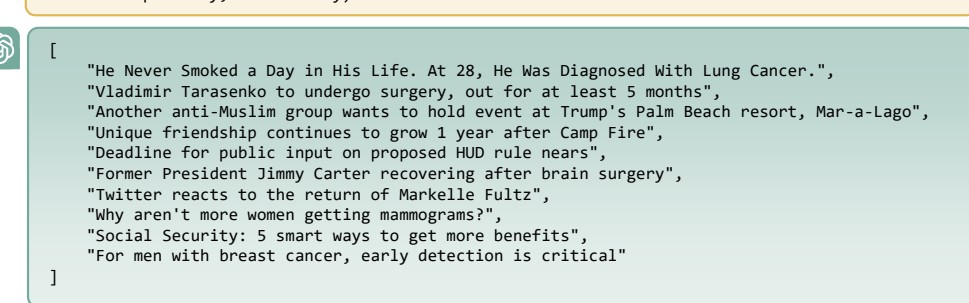

Please return a list of the top **10** most relevant news titles from the unordered candidate pool as a personalized recommendation, based on the user's click history, formatted as a JSON array. Do not include any titles from the user's click history in the recommendations. Please note that the candidate pool is not in any particular order, any ranking based on the order of items in the pool should not be assumed!

**User click history:**
1. The Strangest State Taxes Across America
2. Manchester Town Committee chairman struck, killed by car

…
14. Julian Edelman On Josh Gordon: It's An Unfortunate Situation

**Unordered candidate news title pool (no order should be assumed):**
- Authorities: Dive boat owner faked invoices, photos to collect lionfish bounties
- 18-Year-Old Dead After Accidental Shooting In Fort Smith

…
- Wall Street eyes Washington next week

Top 10 news titles from candidates pool to recommend to the user as JSON array (from highest to lowest priority, titles only):

```
[
    "He Never Smoked a Day in His Life. At 28, He Was Diagnosed With Lung Cancer.",
    "Vladimir Tarasenko to undergo surgery, out for at least 5 months",
    "Another anti-Muslim group wants to hold event at Trump's Palm Beach resort, Mar-a-Lago",
    "Unique friendship continues to grow 1 year after Camp Fire",
    "Deadline for public input on proposed HUD rule nears",
    "Former President Jimmy Carter recovering after brain surgery",
    "Twitter reacts to the return of Markelle Fultz",
    "Why aren't more women getting mammograms?",
    "Social Security: 5 smart ways to get more benefits",
    "For men with breast cancer, early detection is critical"
]
```

(b) Example of Task 2 in Table 14) for MIND

Figure 10: Prompt for MIND

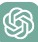

Act as a personalized shopping recommender by analyzing a user's purchase history. Based on
their preferences, recommend the most likely product title they will click from **4** given options.

**User purchase history:**
1. Stripe. Light. Blue. Ladieswear. Blouse in a modal and cotton weave with a round …
2. 2. Solid. Dark. Grey. Lingeries/Tights. Microfibre thong briefs with a low waist, …
…
11. Solid. Dark. Blue. Ladieswear. Top in linen jersey with a V-neck and short sleeves.

**Options:**
A. All over pattern. Dusty Light. Grey. Sport. Ankle-length sports tights in fast-drying …
B. Solid. Medium Dusty. Brown. Ladies Accessories. Suede ankle boots with open toes and …
C. Solid. Light. Yellow. Lingeries/Tights. Push-up bra in lace with underwired, padded …
D. "Lace. Dusty Light. Orange. Divided. Jumper in a soft, fine-knit viscose blend with …

Provide a recommendation from options with no explanation (Format: [A-D]. "Product Title"):

A. All over pattern. Dusty Light. Grey. Sport. Ankle-length sports tights in fast-drying …

(a) Example of Task 1 in Table 14) for HM

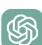

Please return a list of the top **10** most relevant product titles from the unordered candidate
pool as a personalized recommendation, based on the user's purchase history, formatted as a
JSON array. Do not include any titles from the user's purchase history in the recommendations.
Please note that the candidate pool is not in any particular order, any ranking based on the
order of items in the pool should not be assumed!

**User purchase history:**
1. Stripe. Light. Blue. Ladieswear. Blouse in a modal and cotton weave with a round
2. Solid. Dark. Grey. Lingeries/Tights. Microfibre thong briefs with a low waist, …
…
11. Solid. Dark. Blue. Ladieswear. Top in linen jersey with a V-neck and short sleeves.

**Unordered candidate product title pool (no order should be assumed):**
- Stripe. Dark. Green. Ladieswear. Long shirt in woven fabric with a collar, buttons down
- Solid. Bright. Red. Ladieswear. Trousers in woven fabric with a high waist and an …
…
- Solid. Dark. Black. Ladies Accessories. Ankle boots in imitation leather with platform …

Top 10 news titles from candidates pool to recommend to the user as JSON array (from highest
to lowest priority, titles only):

```
[
  "All over pattern. Dusty Light. White. Ladieswear. Blouse in an airy weave with a",
  "All over pattern. Dusty Light. Pink. Baby Sizes 50-98. Two pairs of leggings in",
  "Solid. Bright. Metal. Ladies Accessories. Metal studs in the shape of gift bows.",
  "Solid. Light. Metal. Ladies Accessories. Narrow metal rings in various sizes and designs.",
  "Stripe. Dark. Green. Ladieswear. Long shirt in woven fabric with a collar, buttons down",
  "Solid. Bright. Red. Ladieswear. Trousers in woven fabric with a high waist and an",
  "Solid. Medium Dusty. Brown. Children Accessories, Swimwear. Boots in imitation leather …",
  "All over pattern. Medium Dusty. Brown. Menswear. Boxer shorts in a cotton weave with",
  "Solid. Dusty Light. Orange. Ladies Accessories. Toe-post mules in imitation leather …",
  "Solid. Dark. Khaki green. Divided. Fitted top in glossy jersey with a deep V-neck"
]
```

(b) Example of Task 2 in Table 14) for HM

Figure 11: Prompt for HM

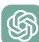

Act as a personalized video recommender by analyzing a user's viewed history. Based on their preferences, recommend the most likely video title they will click from **4** given options.

**User viewed history:**
1. MC Lucky Gun PVP! When you fire a gun in "Desert Ash" and duel with lucky cubes! minecraft
2. Kep1er WADADA Dance Relay Open
…
22. Central Conservatory of Music professors perform famous songs! The roof beautifully …

**Options:**
A. The finished product "hey mama" of the two-hour quick-arrangement of the art candidates …
B. Ultra-short-throw laser TV Epson 735F contrast ratio 2500000: 1 HD restore color visual …
C. Monster Hunter Rise" large expansion "Monster Hunter Rise: Dawn" promotional video
D. Piano x Yuri on Ice | Yuri on ICE

Provide a recommendation from options with no explanation (Format: [A-D]. "Video Title"):

B. Ultra-short-throw laser TV Epson 735F contrast ratio 2500000: 1 HD restore color visual …

(a) Example of Task 1 in Table 14) for Bili

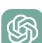

Please return a list of the top **10** most relevant video titles from the unordered candidate pool as a personalized recommendation, based on the user's viewed history, formatted as a JSON array. Do not include any titles from the user's viewed history in the recommendations. Please note that the candidate pool is not in any particular order, any ranking based on the order of items in the pool should not be assumed!

**User viewed history:**
1. MC Lucky Gun PVP! When you fire a gun in "Desert Ash" and duel with lucky cubes! minecraft
2. Kep1er WADADA Dance Relay Open
…
22. Central Conservatory of Music professors perform famous songs! The roof beautifully …

**Unordered candidate video video pool (no order should be assumed):**
- [Times Youth League Ma Jiaqi
- Teach you to make homemade strawberry ice cream, with delicate taste and no ice residue, …
…
- Pea Shooter: There's something wrong with the zombies here today! [These plants are a bit …

Top 10 news titles from candidates pool to recommend to the user as JSON array (from highest to lowest priority, titles only):

```
[
  "Teach you to make homemade strawberry ice cream, with delicate taste and no ice …",
  "[Is this really a human] popping john super control mechanical dance",
  "[Blade of Demon Slayer I Purgatory Kyojuro] Big Brother's 66-Second Heart Challenge",
  "[High-energy Sekiro] The most handsome Iai Kendo, performed in the game!",
  "[Undertale] Stronger Than You Response (ver. Frisk) - Animati",
  "[Hexagonal Palace Lantern]Make it with only a few pieces of paper! I don't buy lanterns …",
  "[Tutorial] Chaoshan Bamboo Oil-Paper Lantern",
  "[Quansheng Dance Studio] Stunning Four ♥ \"Mango\" Chinese Jazz Choreography MV",
  "[Final Fantasy XIV Spring Festival]Gu Raha Tia's Unknown Nursery Rhyme (model …",
  "[Polandball]The country that has been invaded by Germany for the longest time"
]
```

(b) Example of Task 2 in Table 14) for Bili

Figure 12: Prompt for Bili

