# OpenReview forum: "Exploring the Upper Limits of Text-Based Collaborative Filtering Using Large Language Models: Discoveries and Insights"
_ICLR.cc/2024/Conference — Submitted to ICLR 2024_

### Official Review · Reviewer_6KqR · 2023-10-28

**Soundness:** 2 fair
**Presentation:** 2 fair
**Contribution:** 2 fair
**Rating:** 3
**Confidence:** 4

**Summary:**

The paper explores the performance limits and core issues of text-based collaborative filtering (TCF) recommendation models by systematically increasing the size of the text encoder from 100 million to 175 billion parameters. Experiments are conducted using DSSM and SASRec architectures on three datasets.

The results show TCF performance generally improves with larger text encoders, indicating limits have not been reached even at 175 billion parameters. However, fine-tuning on the target dataset remains necessary for optimal performance despite massive pre-trained encoders. Comparisons to ID-based CF reveal that while frozen 175B encoders can achieve competitive results on some datasets, fine-tuning is required to consistently surpass IDCF, especially with DSSM.

Additional experiments demonstrate TCF exhibits some zero-shot transfer ability, but significant gaps remain compared to models adapted to the target data. Overall, the work provides insights into the limits, competitiveness with IDCF, and transferability of TCF models using extremely large language models.

**Strengths:**

1. Systematically studies wide range of encoder sizes up to 175B parameters, revealing performance scaling.
2. Compares TCF to strong IDCF baselines, investigating competitiveness for warm-start recommendation.
3. Examines transfer learning potential, important for general recommender systems.

**Weaknesses:**

1. Only evaluates two basic recommender architectures. More complex models may behave differently.
2. Limited hyperparameter tuning details, so optimal configurations are unclear.
3. Focuses only on random splits, not temporal evaluation protocols.
4. Transfer learning study limited to simple zero-shot approach. More advanced techniques could be explored.

**Questions:**

For transfer learning, have you tried multi-task or explicit domain adaptation techniques?

---

> ### Author Response · Authors · 2023-11-14
> **To Reviewer 6KqR (1)**
>
> We genuinely appreciate your valuable input and the time you dedicated to reviewing our work. However, we have identified certain misunderstandings in your comments. We hope that our sincere answers can win your recognition.  We believe that you, as an open-minded reviewer, will be receptive to our responses and willing to reconsider your score if we can effectively address your key concerns.
>
> >1 Limited hyperparameter tuning details, so optimal configurations are unclear.
>
> We're thinking you might have missed a few places. On page 5, we said “**As for hyper-parameters, we first perform a grid search for standard IDCF as a reference, ..., We report details in Appendix B**” Very detailed hyper-parameter tuning are reported in Appendix B.
>
> >2 Focuses only on random splits, not temporal evaluation protocols.
>
> In page 5, we said the data is split based on temporal orders. See below statement from the original paper:
>
> “The **latest** user-item interaction was used for evaluation, while the **second-to-last** interaction was used for hyper-parameter searching, and all other interactions were used for training”
>
>
> > Transfer learning study limited to simple zero-shot approach. More advanced techniques could be explored.
>
> Zero-shot learning is a significant challenge for the transfer learning research, not only in the domain of Recsys but also in CV and NLP. Evaluating the foundation model's zero-shot prediction ability is crucial, as it sheds light on the progress of our community in terms of transferable or foundation recommendation models. We argue that assessing its performance in this standard zero-shot setting (as in our paper) serves as an essential indicator of our community's current advancements. An important example is CLIP, which highlights its performance under the zero-shot setting.
>
> While we acknowledge the presence of other techniques, such as fine-tuning, adapter tuning, and prompt tuning, it is important to highlight that these parameter tuning approaches necessitate additional computational resources and training examples compared to zero-shot prediction. Zero-shot prediction is a more challenging task in comparison and has more practical values. Although fine-tuning, adapter tuning, and prompt tuning have been acknowledged for their ability to achieve improved performance, the primary objective of this paper is to investigate whether TCF with LLM as the item encoder can accomplish a universal "one-for-all" model. Therefore, evaluating its performance under the fundamental zero-shot setting holds significant importance. Zero-shot prediction vs. finetuning (adapter-tuning or prompt-tuning) is just like the relationship of BERT vs. ChatGPT. No doubt, ChatGPT with zero--shot prediction ability is more appealing and has significant more practical values. It took the NLP community 4 years to move from BERT to ChatGPT.
>
> Please note that our TCF paradigm, as discussed in this paper, is not compatible with some advanced techniques, such as instruction tuning or chain of thought. These techniques are not suitable within the scope of the TCF paradigm because our recommendation backbone is not a LLM. These techniques are mainly developed for LLM. These techniques can be utilized in other recommendation paradigms, such as using LLM as the recommendation backbone, as demonstrated in [1], or based on ChatGPT.  However, we did not focus on such settings in this paper, as LLM-based recommendation models have undergone numerous variations in the past 6 months, among which TCF with LLM as item encoder is an classical and important paradigm.
>
> [1] Recommendation as Language Processing (RLP): A Unified Pretrain, Personalized Prompt & Predict Paradigm (P5)

---

> ### Author Response · Authors · 2023-11-14
> **To Reviewer 6KqR (2)**
>
> > Only evaluates two basic recommender architectures. More complex models may behave differently.
>
>
> We sincerely appreciate your suggestion. In our paper, we acknowledged a limitation of our study as an initial exploration where we only focused on two classical architectures (see Section 4.1). This choice was made because we discovered that even in these well-established architectures, many assumptions do not hold. For instance, even when employing the classical architecture, TCF with LLM still lacks transferability, and items do not possess universality. Given these challenges that persist within classical architectures, we argue that discussing additional recommendation models would not make much sense.
>
> Furthermore, the SASRec model, based on the Transformer architecture, remains the state-of-the-art (SOTA) in recommender systems. The Transformer architecture is not only considered SOTA in the recommender systems community but also in the NLP, CV, and broader AI communities. To date, there is no consensus within the recommender systems community or other AI communities on models that clearly surpass the Transformer.
>
> While many subsequent sequential recommendation models in the recommender systems community compare themselves to SASRec and claim to outperform it, a closer examination of recent papers reveals that SASRec consistently remains the SOTA model, albeit slightly inferior to the proposed model in each respective paper. We believe that the recommender systems community currently faces certain challenges in its development, with one of the most significant issues being the lack of a public benchmark for evaluating the SOTA models. The majority of SOTA models are evaluated based on the authors' own data splits, which raises the possibility that many SOTA models may not truly be the best performers. For example, although BERT4Rec claims to outperform SASRec significantly, subsequent studies have found that BERT4Rec is not as effective as initially claimed. Similar findings can be observed in the comparison between NCF and MF, even though NCF is a more complex model than MF.
>
> In fact, even if some models can surpass SASRec, we believe it would be more of an incremental improvement, as the entire AI community has not made a groundbreaking breakthrough after the basic Transformer architecture. Both GPT and ChatGPT are still based on the Transformer architecture, and certain variations of the Transformer would not affect the core findings of our paper.
>
> The replicability of recommender systems has garnered significant attention in the community in recent years, as many SOTA models have been achieved through unfair means. For example, [2,3,4,5,6,7,8] have highlighted this issue, and a recent study by the authors of GRU4Rec [6] pointed out that the majority of literature with advanced sequential recommendation models did not adhere to the original design when comparing with the basic GRU4Rec, resulting in significantly worse performance compared to the original paper. The pioneers,  Rendle and Koren [2,3],  have also expressed concerns about a significant number of SOTA papers in the recommender systems field over the past decade as they found that once evaluating these SOTA models under fair settings, they were no longer SOTA. Honestly speaking, the reproducibility of research papers in the field of recommender systems have faced catastrophic challenges in  recent  years.
>
> We argue that the observations made on Transformer or SASRec have more general applicability than some SOTA models claimed by the authors. Likewise, the findings for the DSSM model apply to most CTR models as the performance gap between two-tower DSSM and single-tower CTR models in public benchmarks is typically less than 2%, which does not affect our main findings.
>
> [2] Neural collaborative filtering vs. matrix factorization revisited. By Rendle etc. Recsys2020
>
> [3] on the difficulty of evaluating baselines: A study on recommender systems. By Rendle and Koren. 2019
>
> [4]Are We Really Making Much Progress? A Worrying Analysis of Recent Neural Recommendation
>
> Approaches. Recsys2019 Best paper
>
> [5] Reproducibility Analysis of Recommender Systems relying on Visual Features: traps, pitfalls, and countermeasures. Recsys2023
>
> [6] The effect of third party implementations on reproducibility
>
> [7] Our Model Achieves Excellent Performance on MovieLens: What Does it Mean?
>
> [8] Everyone’s a Winner! On Hyperparameter Tuning of Recommendation Models. Recsys2023
>
>
> We hope that our sincere response can alleviate the concerns you may have.

---

> ### Author Response · Authors · 2023-11-14
> **To Reviewer 6KqR (3)**
>
> >Questions: For transfer learning, have you tried multi-task or explicit domain adaptation techniques?
>
> Thanks for this great suggestions. Unfortunately, there is currently no large-scale public dataset available that contains both rich textual data and multiple labels for supporting multi-task learning. While we agree that multi-task learning could potentially lead to more universal representations, achieving zero-shot performance, as discussed in our paper, would require more fundamental advancements in either models or evaluation methodologies.
>
> Regarding explicit domain adaptation, do you mean fine-tuning or retraining the pre-trained model on the target dataset, this approach would lead to comparable or better performance compared to IDRec (see Table 3). However, this is widely recognized as fine-tuning on the target dataset will usually perform no worse than the its training-from-scratch version. However, this is not our focus as performing fine-tuning or other parameter-tuning methods (e.g. adapter tuning or prompt tuning) still requires additional training examples and significant computational resources. The goal of our paper is to explore the possibility of achieving a universal model that can be directly used for downstream tasks without further tuning on new data, similar to. ChatGPT or CLIP etc. This challenge is a crucial one in the fields of NLP, computer vision, and recommender systems. If, in the future, we can develop a universal model for recommender systems, it would significantly reduce the efforts required for individual task-specific recommendation models.
>
> We sincerely hope that our genuine response will meet your approval. If you find our response to address your main concerns, we kindly request that you consider revising the score for our paper. Completing such a paper has costed us $200,000 for over a year. We hope to get some encouragement and recognition from the community.

---

### Official Review · Reviewer_YHSC · 2023-11-01

**Soundness:** 3 good
**Presentation:** 3 good
**Contribution:** 3 good
**Rating:** 8
**Confidence:** 4

**Summary:**

This paper studies the utility of scaling up large language models in the setting of recommendation systems. Recommendation system problems are a fundamentally low-rank and high dimensional problem (the number of missing entries is substantially larger than the number of observed entries) so overfitting is a central issue and integrating neural nets (even simple ones) has not been obvious (as the authors also pointed out, many ideas have been proposed/published but few did fair evaluation). So under the context that building a universal foundational model becomes possible, it is intellectually and practically important to ask how the landscape of building recommender systems is changed. The major questions the authors asked include whether the recommender system also exhibits scaling laws, and whether universal representation is possible. The authors further “partitioned” the questions into smaller, more specific ones that are verifiable/experimentable. The authors’ experiments provide convincing answers to these major questions.

**Strengths:**

The authors asked many natural and important questions related to the interplay between LLM and recommender system; some of the answers are surprising (scaling laws also exist but universal representation is still hard). The execution and experiments are convincing.

**Weaknesses:**

It is a very empirical result and very limited effort is made on the theoretical analysis front.

**Questions:**

Can you comment about the role of overfitting in your work? I noticed people stopped talking about this in neurips/icml/iclr in recent years but the recommender system problems have been closely related to those low rank matrix completion problems, in which significant effort were made to understand variance/bias tradeoff, how the choice on the latent dimensions impact the performance. Is that still relevant when LLM is used for recommender systems (and why/why not relevant)?

---

> ### Author Response · Authors · 2023-11-14
> **To Reviewer YHSC (1)**
>
> Thank you for your valuable review comments and recognition of our work.
>
> > Can you comment about the role of overfitting in your work? I noticed people stopped talking about this in neurips/icml/iclr in recent years but the recommender system problems have been closely related to those low rank matrix completion problems, in which significant effort were made to understand variance/bias tradeoff, how the choice on the latent dimensions impact the performance. Is that still relevant when LLM is used for recommender systems (and why/why not relevant)?
>
> Thanks for raising this question. We agree with you that overfitting is a very important factor when evaluating deep learning models.  In the era of deep learning and LLMs, large models often exhibit strong memorization capabilities, leading to excellent performance on the training set. However, they may suffer from a lack of generalization and perform poorly on the test set or data from different distributions. In recent years, in NLP, CV, and recommender systems communities, training datasets typically reach millions or hundreds of millions of samples, and pre-training data can even reach the billion or trillion level. Consequently, overfitting is not particularly pronounced in these senarios. However, in other fields such as biology, where training samples may be limited to a few hundred, overfitting becomes a significant concern.
>
>
> In our paper, we employed strict data partitioning based on time order. We used the last interaction as the test set, the second-to-last sample as the validation set, and the remaining samples as the training set. This ensures that future user interaction behavior is not included in the training set, as stated in the “Evaluation” section on page 5. Unfortunately, one of our reviewers, 6KqR, said that we randomly split the data and gave us a low score of 3. We hope to receive your support in addressing this issue. Additionally, the same reviewer mentioned that we did not provide details about hyperparameter tuning. However, we clearly state on page 5, "We report hyper-parameter details in Appendix B." Completing a research work is a challenging endeavor, and this work has involved extensive experimentation over a period of more than a year, incurring a substantial cost of approximately $200,000. We hope to receive a fair and impartial review. We kindly request your assistance in advocating for our work.

---

> ### Author Response · Authors · 2023-11-14
> **To Reviewer YHSC (2)**
>
> >your subquestion: how the choice on the latent dimensions impact the performance. Is that still relevant when LLM is used for recommender systems (and why/why not relevant)?
>
> In general, increasing the embedding dimension tends to improve performance until it reaches a point of stability. The optimal embedding size varies depending on the dataset and typically falls within the range of {100, 1000}. For example, in our Appendix B, we rigorously searched for the optimal embedding dimension among {64, 128, 256, 512, 1024} and determined the best dimension through experimentation. Unfortunately, many papers often overlook fine-tuning relevant hyperparameters, especially the embedding dimension. This can easily lead to self-proclaimed SOTA results. However, our findings demonstrate that even with a 175B LLM, its output representations only achieve comparable performance to the basic ID embeddings. We believe that fair comparisons are crucial. Without fair comparisons, it becomes difficult to determine the source of performance improvements. When comparing two models that use different loss functions and samplers, the results can be significantly influenced, making it challenging to ascertain if the claimed performance improvement is valid.
>
> Regarding LLM research, there are generally two approaches. One is to use the LLM as the item encoder, replacing the itemID embedding with the LLM's representations. The other approach is to employ the LLM as the recommendation backbone, as shown in Appendix Figure 7 of our paper. However, we found that the latter approach significantly lags behind the paradigm we discussed, which is utilizing the LLM as the item encoder. Besides, using LLM as the recommendation backbone is much more expensive than our discussed TCF paradigm. In our TCF paradigm, the item representation can be pre-extracted from the LLM，and then add them as common fixed features for recommendation models. However, if we use LLM as backbone, it will be extremely expensive as 1) the sequence becomes much longer (the number of items times the number of words per item); 2) the recommendation backbone becomes much larger (deeper and wider) than commonly used backbone network, such as SASRec (SASRec just deals with items, while this paradigm deals with words, and requires a larger network to fill the capacity, like other foundation models in NLP). In this paper, we mainly focus on the TCF paradigm, which is a very common setting.
>
> > It is a very empirical result and very limited effort is made on the theoretical analysis front.
>
> Thank you for raising this question. We acknowledge that this work lacks some theoretical analysis, as we find that some issues may not be well theoretically analyzed.  Instead, we have conducted extensive empirical studies, which are resource-intensive. For example, in Section 6, we conducted fine-tuning of the LLM with a model size of tens of billions of parameters. This process necessitated the use of dozens of A100 80G GPUs and significant time investments for each experiment. As a result, the compute costs for these experiments exceeded $200,000. We realize such expensive experiments are not commonly seen in the recommender system field. We find that  top-tier conferences like ICLR, NIPS and CVPR have published quite a few pure empirical research papers, such as "chain of thought"[1] and literature [2,3,4,5,6,7]. We fully agree with you with some theoretical analysis will make the paper stronger.
>
> 1) Chain-of-Thought Prompting Elicits Reasoning in Large Language Models. NeurIPS2022
>
> 2) Rethinking ImageNet Pre-training. By Kaiming He etc. CVPR 2019
>
> 3) Bag of Tricks for Image Classification with Convolutional Neural Networks. CVPR2019
>
> 4) An Empirical Study of Training Self-Supervised Vision Transformers. CVPR2020
>
> 5) Emergent Abilities of Large Language Models. Transactions on Machine Learning Research2022
>
> 6) Exploring the Limits of Transfer Learning with a Unified Text-to-Text Transformer. Journal on Machine Learning Research2019
>
> 7) AN EMPIRICAL STUDY OF EXAMPLE FORGETTING DURING DEEP NEURAL NETWORK LEARNING. ICLR2019
>
> We genuinely appreciate your valuable feedback and greatly appreciate your support. We humbly request your further support when discussing this paper with other reviewers and the AC.

---

### Official Review · Reviewer_6ddN · 2023-11-01

**Soundness:** 3 good
**Presentation:** 4 excellent
**Contribution:** 3 good
**Rating:** 6
**Confidence:** 3

**Summary:**

This paper studies the impact of using large language models as text encoders for text-based collaborative filtering on the recommendation performance. It does not propose a new method, but conducts extensive experimental research on the effect of text encoders of different parameter scales on TCF algorithms. The main conclusions are as follows:
1. Larger parameter text encoders can continuously improve the performance of TCF, but even at the scale of the OPT-175B model, it is impossible to achieve the ideal universal representation and is still weaker than the fine-tuned small models.
2. For recommendation scenarios where the main features of the item are text, TCF on the 175B model can achieve a similar performance to the IDCF algorithm and can remove the item ID feature without losing the recommendation effect.
3. Even with the universal item ID feature, due to the possible differences in the matching relationship between users and items in different recommendation applications, the performance of directly transferring the matching model of a certain domain to other domains in a zero-shot manner is still poor.

**Strengths:**

1. The paper investigates the role of LLM in constructing a universal and transferable recommendation system for text-based collaborative filtering, conducts very extensive experiments, and the work is novel and interesting.
2. The experimental results show that for text-centric recommendation applications, the TCF on the OPT-175B model can achieve comparative performance to standard IDCF algorithm, which is enlightening for the construction of a universal and transferable recommendation system based on LLM.
3. The paper is well-organized, well-written, and easy to understand.

**Weaknesses:**

1. The experimental datasets use the title as the item feature, and there may be more information that can be utilized but has not been used, leading to the potential of the tested method being underestimated.
2. The paper is mainly experimental and does not propose new solutions.

**Questions:**

Please address the issues mentioned in the Weaknesses section.

---

> ### Author Response · Authors · 2023-11-14
> **To reviewer 6ddN**
>
> Thank you for your valuable review comments and recognition of our work.
>
> > The experimental datasets use the title as the item feature, and there may be more information that can be utilized but has not been used, leading to the potential of the tested method being underestimated.
>
> This is a very good question, and indeed, as you suggested, using more information could potentially improve the performance of TCF. For example, in the case of news recommendations, we could utilize both the title and the full text content. We are now conducting related experiments and will report these results later.
>
>
> We did not do this in the original submission primarily due to the computational challenges involved in using full-length texts. Many experiments with full-length texts would require ultra-high computational resources, such as the fine-tuning experiments and experiments on LLMs with billions of parameters.
>
> We have just completed the experiment where we have taken into account both the abstract and the title of a new article. Please find the updated results presented below.
>
> > The paper is mainly experimental and does not propose new solutions.
>
> We greatly appreciate your comment. As you mentioned, this paper does not aim to propose new deep learning techniques. Instead, its objective is to shed light on the current advancements within the recommender systems community, specifically focusing on LLM4Rec or TCF‘s scaling effect, and to explore the possibility of establishing a foundation recommendation model akin to BERT or ChatGPT in the field of natural language processing (NLP).
>
> We recognize that recommender systems still have a considerable journey ahead to catch up with NLP and computer vision (CV) in terms of large-scale foundation models. Presently, the recommender systems community lacks readily available one-for-all  pre-trained models (e.g. in HuggingFace)  that can be applied to various downstream tasks, similar to NLP's BERT. This gap signifies that developing a general-purpose model in the realm of recommender systems might pose more significant challenges compared to NLP and CV.
>
>
> Second this paper demonstrates some surprising findings, for example, even with a 175B LLM, the results of TCF are only comparable to those IDRec by using simple itemID embedding features. This finding is unexpected because numerous papers claim to have achieved state-of-the-art (SOTA) performance with their multimodal or text-based recommendation models. However, upon reviewing the related literature, we discovered that these papers often rely on unfair comparison settings, such as using different backbones, loss functions, and samplers for comparison. Such comparisons make it challenging to attribute improvements solely to LLMs or other components.
>
> We feel that empirical studies are vital for the recommender systems community. We have noticed that ICLR receives a significant number of NLP and CV papers each year that focus on empirical studies. By contrast, there are relatively few empirical studies in the field of recommender systems published at ICLR.  Additionally, while this paper is an empirical study, it involves extensive and expensive experiments. For example, fine-tuning the billion-parameter LLM requires dozens of A100 GPUs, resulting in compute costs exceeding $200,000. We hope that our extensive experiments will provide the community with some useful insights. In fact, the fields of NLP and CV have seen numerous empirical study papers, and it would be unfortunate if the recommender systems community hindered such research. Therefore, we sincerely hope to receive your continued support.
>
> We listed several high-impact papers which are purely empirical and published in top-tier venues e.g. CVPR and ICLR.
>
> 1) Chain-of-Thought Prompting Elicits Reasoning in Large Language Models. NeurIPS2022
>
> 2) Rethinking ImageNet Pre-training. By Kaiming He etc. CVPR 2019
>
> 3) Neural Collaborative Filtering vs. Matrix Factorization Revisited. Recsys2020
>
> 4) An Empirical Study of Training Self-Supervised Vision Transformers. CVPR2020
>
> 5) Emergent Abilities of Large Language Models. Transactions on Machine Learning Research
>
> 6) Exploring the Limits of Transfer Learning with a Unified Text-to-Text Transformer. Journal on Machine Learning Research
>
> 7) AN EMPIRICAL STUDY OF EXAMPLE FORGETTING DURING DEEP NEURAL NETWORK LEARNING. ICLR2019
>
> 8) Bag of Tricks for Image Classification with Convolutional Neural Networks. CVPR2019
>
> Finally, please allow us to express our gratitude for your support and great advice.

---

> ### Author Response · Authors · 2023-11-16
> **To reviewer 6ddN**
>
> Thanks for your constructive feedback. We have just conducted the experiments as you advised, considering additional text on the MIND dataset. The results are reported on the below table regarding HR@10. Based on the results, we have observed that adding more text does not necessarily result in improved performance. Kindly note that that the other two datasets we used only consist of titles.  All hyper-parameters are kept the same for comparison purpose.
>
>
> | Model Size | Title & Abstract | Title Only |
> |------------|-------------------|------------|
> | 125m       | 19.30            | 19.07    |
> | 66b        | 19.84             | 20.11    |
> | 175b       | 20.00           | 20.24    |
>
>
> We genuinely appreciate your valuable feedback and humbly request your further support when discussing this paper with other reviewers and the AC.

---

### Official Review · Reviewer_Jj3E · 2023-11-04

**Soundness:** 3 good
**Presentation:** 3 good
**Contribution:** 3 good
**Rating:** 5
**Confidence:** 4

**Summary:**

In this paper, the authors perform empirical experiments to analyse the upper limits of the text-based collaborative filtering (TCF) recommendation systems. They progressively increase the size of item encoders from one hundred million to one hundred billion to reveal the scaling limits of the TCF paradigm. Moreover, they also study whether the extremely large LLMs can enable a universal item representation for recommendation task. The analysis presented in this work not only demonstrates positive results but also uncovers unexpected negative outcomes, showing the current state of the TCF paradigm within the community.

**Strengths:**

1. This paper presents an empirical study analysing the performance limit of existing TCF paradigm.

2. The authors study the scaling limits of the TCF paradigm, and also investigate the extremely large LLMs can enable a universal item representation for the recommendation tasks.

3. This study not only shows positive results but also shows unexpected negative results, showing the current state of TCF paradigm within the community. The findings introduced in this paper may inspire further research on the text-based recommender systems.

**Weaknesses:**

1. The studied problem is interesting and very important for recommender system research, and the experimental results may also inspire future research in recommendation systems. However, this paper may not be suitable to ICLR, it should be better to submit this paper to the IR conferences or journals.

2. This paper only show the empirical experimental results and present relevant discussions. It does not introduce some novel deep learning techniques.

3. Some details of the experimental settings are not clear. In the experiments, the authors also fine-tune the LLMs with the data in recommendation domains. However, it is not clear what kind of data are used in LLM fine-tuning, and how to fine-tune the LLMs.

**Questions:**

1. For sequential recommendation, there are some SOTA models that have more complex structure than SASRec and DSSM. Why not using these SOTA sequential recommendation models with more complex structures as backbone models to study the TCF?

2. What kind of data in recommendation domain are used to fine-tune the LLMs and how to fine-tune the LLMs?

3. According to my understanding, the TCF methods studied in this paper use the LLM as the item encoder and use traditional sequential recommendation models to model users' behaviours. Is it possible to directly use LLMs to model the user behaviours?

---

> ### Author Response · Authors · 2023-11-14
> **To reviewer Jj3E (1)**
>
> Thank you for your valuable review comments. We will make our best effort to address your questions and concerns, and we hope to earn your approval.
>
> >The studied problem is interesting and very important for recommender system research, and the experimental results may also inspire future research in recommendation systems. However, this paper may not be suitable to ICLR, it should be better to submit this paper to the IR conferences or journals.
>
> We appreciate your question, and we hope that our sincere response can address your concerns. In this paper, we study LLM representation for recommender systems, specifically by discussing scaling, representation university, transfer learning to investigate whether very large language models can revolutionize the field of recommender systems. These research questions align with the domains of representation learning and transfer learning, which are within the scope of the International Conference on Learning Representations (ICLR).
>
>
> We listed three research topics from calling for paper of ICLR2023:
>
> 1)"**Representation learning for computer vision, audio, language, and other modalities**"
>
> 2)"**Transfer Learning, Meta-Learning and Lifelong Learning**"
>
> 3)"**interpretation of learned representations**"
>
> > This paper only show the empirical experimental results and present relevant discussions. It does not introduce some novel deep learning techniques.
>
> We appreciate your concern. We have realized that there are quite a few empirical study papers in every year‘s ICLR proceedings, as well as a significant number of related paper in CVPR and ACL, e.g.[1,2,3,4,5,13,14].  Many of them have very high impact on their research communities. Empirical study papers, unlike typical algorithm-focused papers, usually do not necessarily require the introduction of new deep learning techniques. However, they often delve deeper into important or common research paradigms or deep learning models, providing valuable insights and guidance to the community. These papers can stimulate further reflection and inspire future research. We believe that high-impact research should initiate community discussions and reflections, highlighting existing shortcomings and challenges. Papers, such as empirical studies, rethinking studies, datasets, and benchmarks, often focus on studying existing deep learning techniques rather than introducing new ones. Another example is that the call for paper part also mentioned one research direction is "visualization or interpretation of learned representations". Our paper can be seen as an interpretation of learned representation of LLM for the recommender system problem.
>
>
> **We feel that introducing new deep learning techniques is not the sole criterion for a good research paper in the field of AI. Instead, we believe that the most significant papers are those that uncover important yet unknown facts, inspire and motivate the community, and stimulate new discussions.  If these important findings have not been revealed before, they should be regarded as novel.  Identifying problems can sometimes be as important as solving them. In the recommender systems community, we believe it is also important to encourage diverse research.**
>
> >Some details of the experimental settings are not clear. In the experiments, the authors also fine-tune the LLMs with the data in recommendation domains. However, it is not clear what kind of data are used in LLM fine-tuning, and how to fine-tune the LLMs.
>
> In section 6, we conduct experiments involving fine-tuning of LLMs. We utilize the training dataset mentioned in Table1  to perform the fine-tuning process. The key distinction in these experiments compared to other frozen LLM experiments is that the LLMs, when used as item encoders, undergo optimization during the training process, whereas frozen LLMs maintain their LLM parameters unchanged. Also see description on page 7 around footnote 8 and figure 3.
>
>
> [1] Chain-of-Thought Prompting Elicits Reasoning in Large Language Models. NeurIPS2022
>
> [2] Rethinking ImageNet Pre-training. By Kaiming He etc. CVPR 2019
>
> [3] Neural Collaborative Filtering vs. Matrix Factorization Revisited. Recsys2020
>
> [4]An Empirical Study of Training Self-Supervised Vision Transformers. CVPR2020
>
> [5]AN EMPIRICAL STUDY OF EXAMPLE FORGETTING DURING DEEP NEURAL NETWORK LEARNING. ICLR2019
>
> [13] Emergent Abilities of Large Language Models. Transactions on Machine Learning Research2022
>
> [14] Exploring the Limits of Transfer Learning with a Unified Text-to-Text Transformer. Journal on Machine Learning Research2019

---

> ### Author Response · Authors · 2023-11-14
> **To reviewer Jj3E (2)**
>
> Questions:
>
> > For sequential recommendation, there are some SOTA models that have more complex structure than SASRec and DSSM. Why not using these SOTA sequential recommendation models with more complex structures as backbone models to study the TCF?
>
> DSSM and SASRec (with Transformer as the backbone) are regarded as among the most representative models in the field of recommender system. DSSM can represent the performance of the majority of CTR models, as the main difference lies in the single-tower versus dual-tower architecture, which has minimal impact on performance and does not affect the core findings of our paper. The difference between single towers and twin towers usually fluctuates within 2%.
>
> While some papers claim that their new models outperform SASRec, a closer look at their future work reveals that SASRec consistently remains the top baseline, usually only below the model proposed by the authors. It is important to note that some SOTA improvements are not necessarily due to the claimed advancements proposed by the authors. Often, they arise from unfair comparisons, such as using different loss functions, samplers, or suboptimal hyperparameters. We have extensively evaluated numerous SOTA models proposed that claimed better than SASRec, but unfortunately, none of them (e.g., BERT4Rec, P5[10], SSE-PT[11], etc) have really surpassed SASRec's performance by properly tuning the hyper-parameters.  There are too many SOTA papers that do not clearly show their evaluation details (we realized that many papers compare baselines at randomly chosen embedding sizes (e.g. 100)), which is not convincing as a SOTA claim should be based on carefully tuned optimal embeddings and hyper-parameters. Some improvements will disappear if we enlarge the embedding size or model size of baseline models. Consequently, these claimed SOTAs may only represent the authors' own SOTA rather than being widely accepted as community-wide SOTAs.)
>
>
>
>
> In recent years, the reproducibility of research papers in the field of recommender systems have faced catastrophic challenges. Numerous subsequent papers have pointed out that the credibility of SOTA (State-of-the-Art) models in recommender systems is highly questionable, see [6,7,8,9]. The paper [9] highlights that there are hardly any correct third-party implementations of GRU4Rec (all of these well-known implementation is much weaker than the authors' official implementation), and Rendle and Koren [6,7] has also expressed serious doubts about a significant number of so-called SOTA papers in the recommender systems field over the past decade. Therefore, we believe that studying the problem using the most widely recognized baselines is more convincing and representative than relying solely on self-proclaimed SOTA models.
>
> In addition, even assume there are some more advanced models, we believe they are only incremental improvements because so far there are no groundbreaking networks after Transformer, neither in Recys nor in NLP and CV. These new variants or incremental improvement of DSSM and Transformer intuitively will not affect the core findings in this paper. We agree with our reviewers, evaluating more models will be interesting, but it is very difficult to present too many results and discussions due to a limited 9 page space.  We argue that a good paper is the one inspires more research but not the one solving all problems or considering all aspects. In addition, we have shown some unexpected or surprising results on the two classical models and if we cannot solve them on the standard models, it seems not very meaningful to study many other variants at this stage.  To sum up, Transformer or SASRec is still a SOTA model in terms of network architecture in the recsys community and the broader AI community.  These incremental improvements or variant models will not have a significant impact on our findings. In fact, many empirical study work in NLP and CV are also evaluated on the Transformer architecture as so far Transformer still represents the classical and SOTA network backbone in these fields (as well as in Recsys).
>
>
> [6]Neural collaborative filtering vs. matrix factorization revisited. By Rendle etc. Recsys2020
>
> [7] on the difficulty of evaluating baselines: A study on recommender systems. By Rendle and Koren. 2019
>
> [8] Are We Really Making Much Progress? A Worrying Analysis of Recent Neural Recommendation Approaches. Recsys2019 Best paper
>
> [9] The effect of third party implementations on reproducibility.  Recys2023
>
> [10] Recommendation as Language Processing (RLP): A Unified Pretrain, Personalized Prompt & Predict Paradigm (P5)
>
> [11] SSE-PT: Sequential recommendation via personalized transformer. (They used a larger embedding size for their model during comparison, if we tune the embedding properly, the improvement does not exist.)

---

> ### Author Response · Authors · 2023-11-14
> **To reviewer Jj3E (3)**
>
> > What kind of data in recommendation domain are used to fine-tune the LLMs and how to fine-tune the LLMs?
>
> The data set is shown in Table 1. We use the recommended dataset to train the model, the only difference is that when we talk about fine-tuning, the LLM is trained along with parameter updates. Otherwise, LLM will be frozen. Fine-tuning is done in the same way as retraining the model, but using a pre-trained LLM as initialization for the item encoder.
>
> >According to my understanding, the TCF methods studied in this paper use the LLM as the item encoder and use traditional sequential recommendation models to model users' behaviours. Is it possible to directly use LLMs to model the user behaviours?
>
> This is a very good question. This paper we focus on LLM as item encoder, but we also evaluate using LLM as the backbone to model user behavior as reported in Appendix D figure 7. Unfortunately, we did not find using LLM as user backbone shows satisfied results. This architecture is very very costly in terms of compute and memory compared to using LLM as item encoder. Assume that we have 100 items, and each item has 100 words, the sequence is up to 10, 000, very challenging to train such big models and perform online inference.
>
>
> We believe that you are an open-minded and receptive reviewer, and we are committed to addressing your questions to the best of our abilities. We kindly request your support if we can resolve your main concern. Although our work is an empirical study, we have made highly novel discoveries that are  important to the community. To conduct the relevant experiments, we have invested nearly $200,000, which is uncommon for research papers in the recommender systems community. We hope to receive the support of the reviewers. **The fields of NLP and CV have seen numerous empirical study papers, and it would be unfortunate for the recommender systems community to stifle such research. In fact, in recent years, particularly during the deep learning era, the recommender systems community has lagged behind NLP and CV communities. Our community relies on you!**
>
> Thank you once again for your diligent review of our paper. We genuinely hope that our sincere response has met your approval. If you find our response satisfactory in addressing your main concerns, we kindly request that you consider raising the score for our paper. Completing this paper has required a significant investment of $200,000 over the span of a year. We earnestly hope to receive encouragement and recognition from the community.

---

### Author Response · Authors · 2023-11-22
**To  AC - no reply from reviewers with such a discrepant score**

We apologize for the disruption, but we would like to express our concerns regarding the rebuttal process. So far, we have not received any response from the two reviewers who gave low scores. Our paper has received a score of 8653, which indicates a significant discrepancy. Moreover, we have clearly pointed out some evident errors in the feedback provided by the reviewers.

In fact, last year we also submitted a Recsys paper to ICLR and received a score of 86553. Unfortunately, the reviewer who gave a score of 553 did not provide any response even after a month of rebuttal. Is this the norm in the recommender systems community? We are deeply disappointed with the review ecosystem of recommender systems in recent years.

We genuinely hope that our AC will take a bit time to read our paper and assess the comments from the reviewers. We do not agree with the reviewers' statement that our paper LLM4Rec is not suitable for ICLR and that empirical study research is a weakness. We believe that this reflects the reviewers' own limitations and perspectives. It is highly unprofessional for the ICLR reviewers to claim that empirical research must introduce new deep learning techniques.

Thanks

Authors

---

### Author Response · Authors · 2023-11-23
**To all reviewers**

Dear all reviewers,

We sincerely appreciate the feedback and questions provided by all the reviewers. We have addressed these core concerns raised and hope that our sincere responses will earn your approval. We have noticed that there might be some misunderstandings regarding certain issues, and we hope that our responses can address your concerns effectively.

Authors of paper 1750

---

### Public Comment · ~FengyiLi1 · 2023-12-03
**To reviewer Jj3E**

Investigating models with hundreds of billions of parameters is a non-trivial task that demands substantial computational resources, time, and engineering efforts. Google has published numerous empirical study papers focusing on NLP problems (e.g. scaling law of LLMs) in top-tier conferences and journals. Similarly, in the field of recommender systems, we should encourage and support diverse research approaches.

---

> ### Public Comment · ~FengyiLi1 · 2023-12-05
> **Regarding what is the significance of the work?**
>
> Regarding what is the significance of the work? Empirical study can clearly be a significant contribution for the ICLR community. This is copied from the ICLR official website for reviewer guide：https://iclr.cc/Conferences/2024/ReviewerGuide
>
> “Does it contribute new knowledge and sufficient value to the community? Note, this does not necessarily require state-of-the-art results. Submissions bring value to the ICLR community when they convincingly demonstrate new, relevant, impactful knowledge (incl., **empirical, theoretical**, for practitioners, etc).”
>
> In addition, to determine if a submission is a good fit for ICLR, it is advisable to refer to the call for papers of ICLR. The conference focuses on topics such as representation learning, transfer learning, related applications, and benchmarks. These areas fall within the scope of ICLR: https://iclr.cc/Conferences/2024/CallForPapers

---

### Public Comment · ~FengyiLi1 · 2023-12-08
**To Ac**

As reviewers, it is their duty to select good papers for the community. Regrettably, there are instances where some reviewers do not approach the reviewing process with the seriousness, as evidenced by both authors' feedback and public comments. Despite the authors' clear explanations addressing the reviewer's incorrect comments, the reviewer has chosen not to respond.

---

> ### Public Comment · ~FengyiLi1 · 2023-12-10
> **Hope AC treat the reviewers' comments fairly**
>
> The authors spent 2 weeks and tried their best to clarify some obvious mistakes made by the reviewer 6KqR, but there was no reply. Hope AC and SAC can evaluate the paper and reviewers' comments fairly.

---

> ### Public Comment · ~FengyiLi1 · 2023-12-10
> **Hope AC treat the reviewers' comments fairly**
>
> The authors spent 2 weeks and tried their best to clarify some obvious mistakes made by the reviewer 6KqR, but there was no reply. Hope AC and SAC can evaluate the paper and reviewers' comments fairly.

---

### Public Comment · ~Angelinthesun1 · 2023-12-11
**To reviewers**

To be  a qualified reviewer, we probably need to read “How to be a good CVPR Reviewer”: https://go.iu.edu/4Ruz. They are applicable to ICLR reviewers as well and can be very useful.

The prosperity of the computer vision community can be attributed to the presence of reliable reviewers and ACs. This also applies to the Recsys community.

Setting aside this specific paper, it is evident that certain opinions by reviewers are untenable. For instance, **reviewer Jj3E's claim that empirical study papers without introducing new deep learning techniques are a weakness, which we believe ICLR has no such Weakness policy.**  Additionally, his another assertion that this paper does not align with the scope of ICLR does not hold as well. The paper investigates the impact of large language models (LLMs) on the field of recommender systems clearly aligns with the scope of the call for paper in ICLR. Why it is not suitable to ICLR? The reviewers' comments are not properly justified! **In the CVPR reviewer guidelines, it is stated, "CVPR is very inclusive: Historically, rejection solely for out-of-scope is rather rare."** I believe ICLR is also very inclusive!

As for reviewer 6KqR, some comments intentionally misrepresent the objective description of the paper. During the two-week rebuttal process, despite multiple reminders from the authors and ACs, he consistently refused to respond. This attitude of knowingly holding erroneous review opinions but refusing to correct them cannot be accepted!


**As reviewers, please do not let researchers in your community down. Reviewers and authors should respect each other's effort.**


This is copied from CVPR reviewer guideline: https://go.iu.edu/4Ruz

    lf you write bad, poorly justified, ill-considered or unfair reviews....

    Area and Program Chairs, who may greatly influence your career advancement, may remember that you let them down.

    Authors may feel unwelcome or mistreated by the review process.

    A reader may waste time on a flawed or uninformative paper that wasaccepted, or may waste time in research because a valuable paper wasrejected.

    lf you write good, insightful, well-justified,constructive reviews...

    Area and Program Chairs will love you because you will make the paperdecision much easier.

    The authors' faith in the vision community will increase, and, even ifthey need to resubmit, they will know what needs to improve.

    Researchers will continue to flock to vision conferences for the latest anogreatest in computer vision ideas and techniques.

    AC/SACs know your names! They will not have a good impression of you if you submit sloppy or late reviews

    Be aware of your own bias. We all tend to assign more value to papers that are relevant to our own research.

    Reviewer: You must have a theorem! No such policy

    Reviewer: You must beat SOTA! No such policy


"**Try to ignore interestingness of topic or “fit to the conference" and focus on whether the paper can teach something new to an interested reader.**"

"**Make your final recommendations with solid justifications. Read the rebuttal and discussions. Do they change your position? Why?**"  also from CVPR reviewer guideline.

Generally speaking, the opinions of the two reviewers who gave low scores lack factual grounds and their review opinions are unconvincing！

---

### Public Comment · ~Public_Chair1 · 2023-12-11
**Please stop abusing public comments!**

To authors,

Please stop abusing public comments.
Reviewers/AC may change their judgments due to authors' unprofessional behavior using sock puppet accounts.

---

> ### Public Comment · ~Angelinthesun1 · 2023-12-11
> **Hi there**
>
> Openreview allows everyone to express a reasonable evaluation of low-quality review comments
>
> This paper  received conflicting scores (8653), covering almost all score ranges. In such cases, ICLR strongly recommends that reviewers who gave lower scores should engage with the authors at least once to assess whether the authors' response adequately addresses their concerns. This interaction is considered a basic requirement for a reviewer, as it is the purpose of the rebuttal process in ICLR. Reviewers are aware of and agree to this expectation when they accept the role of reviewing for ICLR.
>
> Authors dedicated two weeks to addressing the reviewers' questions, but reviewers Jj3E and 6KqR remained silent, unwilling even to say I had read the rebuttal. Is this behavior considered respectful towards the authors' efforts?
>
> From an author's perspective, this is not respectful!  **I firmly believe that it is not the authors' fault to expect reviewers to at least provide a response, indicating that they have read the rebuttal. Is this an excessive expectation?  I believe reviewers  will not keep silent if their own paper is mistreated.**
>
> The purpose of OpenReview is to make the peer review process more transparent and fair, so it is reasonable for incompetent reviewers to receive criticism through public comments.

---

> ### Public Comment · ~Feng_Cheng2 · 2023-12-12
> **comments from a random person**
>
> I checked the recent activities and randomly found this thread.
> I noticed there are a large number of papers with no responses from reviewers after rebuttal. Even with responses, some reviewers only change the rates without leaving any comments or rationales for the adjustments.
>
> While I quite like ICLR's rebuttal format, I am disappointed with the overall review quality of ICLR this year.

---

> ### Public Comment · ~Angelinthesun1 · 2023-12-13
> **There is nothing we can do!**
>
> This is not an issue specific to ICLR; it is a problem within the recommender systems community itself. In other communities, when controversial papers arise, there is often active engagement and discussion. However, we are uncertain about the situation within the recommender systems community, as the lack of interaction renders the rebuttal process meaningless.

---

### Meta-Review · Area_Chair_jHtM · 2023-12-10

**Metareview:**

Although it is a borderline paper from reviewers' feedback, I am inclined to reject this paper after reading this paper myself. This paper is an empirical study to explore the capability of LLM-powered text-based item encoders to improve two popular text-based collaborative filtering (CF) methods: DSSM and SASRec with transformer as the backbone compared to the ID-based CF. There are some interesting findings in this study with OPT-175B. However, many of the high-level insights about the TCF and IDCF are similar to an earlier work [1]. The major difference in the experimental setup is that the "LLM" in [1] is BERT and RoBERTa and the "LLM" in this paper is OPT-175B. Even the two CF methods (DSSM and SASRec) included in this paper are the same ones included in [1]. As a result, the findings in this paper seem to be incremental and the way to conduct empirical study is also not novel. As a result, I recommend to reject this paper.

[1] Yuan et al. Where to go next for recommender systems? id-vs. modality-based recommender models revisited. SIGIR 2023

**Justification For Why Not Higher Score:**

Please see the "Additional Comments On Reviewer Discussion".

**Justification For Why Not Lower Score:**

N/A

---

> ### Public Comment · ~Angelinthesun1 · 2024-04-23
> **Some basic clarification to meta review**
>
> The final decision and some of the reviewer feedback are disappointing. The paper received divergent feedback (8653) from reviewers initially, indicating that some reviewers may have been incorrect in their assessment. In such cases, reviewers and AC should engage in a dialogue with the authors after reading their rebuttal. Unfortunately, there was no any interaction from the reviewers, rendering the two-week rebuttal phase meaningless.  As ICLR reviewers, they should have basic interaction with the authors when there are divergent opinions. The AC should remind reviewers about this requirement. What's the point of rebuttal if all the reviewers are silent.
>
> There are some clarification on the meta review:
>
> >Although it is a borderline paper from reviewers' feedback.
>
> Papers with widely varying scores are generally not considered truly borderline.This usually means that some reviewers may be wrong or unqualified to serve as reviewers for this work. In such cases, the AC is encouraged to carefully review all the comments from the reviewers and the authors' rebuttal. It is clear that Reviewer 6KqR's reviews contain key factual errors, and the AC should ask the reviewer to update their comment. A paper with a score for example, 5566, is a borderline paper.
>
> >However, many of the high-level insights about the TCF and IDCF are similar to an earlier work [1].
>
> The paper has adequately clarified the differences from [1] in several places. It seems that the AC did not thoroughly read this part or perhaps did not grasp the key points. The paper presents fundamentally different findings from [1]. One crucial distinction is that all the findings in [1] are based on end-to-end training, meaning that LLMl and recommender backbone need to be trained jointly to achieve state-of-the-art (SOTA) results. However, as [1] points out, end-to-end training requires over 100 times more computation and training time, making it impractical for large-scale industrial systems. Moreover, the paper demonstrates that pre-extracted text features perform poorly compared to the basic IDCF. However, in this paper, we show that with a 100B LLM, TCF can be comparable to IDCF even with pre-extracted textual features, which represents significant progress. Unfortunately, this seems to have been overlooked by the AC.
>
> Furthermore, the increase in LLM from 100 million to 100 billion should not be considered as incremental research. The AC seems to be unaware of the significant challenges involved in conducting research from 100 million to a 100B LLM.
>
> While scaling LLM has been extensively studied in NLP, this is the first paper to investigate LLM scaling for recommender systems. For instance, without conducting extensive experiments, no previous research (including [1]) was aware that **a 100B LLM is still not universal, not transferable for the Recsys task, and only achieves comparability to the basic IDCF.**  These findings are completely new as no any paper demonstrated them. Note [1] does not investigated the transferability of LLM and does not study any LM over 1B. The key findings are different. Research studying LLM with 1000x larger model size  with so many new findings should not be regarded incremental.
>
> It appears that some reviewers and the AC did not recognize this, with one reviewer even suggesting that this was merely an empirical study without a technical component, and therefore should not be accepted by ICLR. It is important to note that ICLR does not require every accepted paper to propose a new method, and empirical studies hold significant value in research.
>
> Lastly, it is important to acknowledge that it is impossible to address all problems and cover all aspects in a single paper. The paper focuses on studying two well-known backbones, DSSM and Transformer. The inclusion of Transformer is justified as it remains the state-of-the-art model not only in the field of Recsys but also in the broader AI community. Similarly, DSSM serves as a fundamental deep learning model for Recsys. As explained in the paper, even if DSSM is replaced with a single tower model, the differences in performance would not be substantial, as numerous studies have shown that the improvement from DSSM to a single tower model is only marginal (usually less than 2% under a fair comparison). By considering the findings presented in this paper, one can easily infer that the key discoveries will remain consistent even when experiments are conducted using other single tower models such as DeepFM or wide & deep. This aspect should not be considered a significant weakness by an expert reviewer.

---

> ### Public Comment · ~Angelinthesun1 · 2024-04-23
> **Key difference from literature [1]**
>
> This work was indeed inspired by [1], but it has made several significant contributions.
>
> First, [1] primarily focused on smaller LMs such as BERT and RoBERTa, which had parameter sizes around 100 million. In contrast, our research investigates the scaling effect of super large LMs ranging from 100 million to 175 billion parameters. Extensively studying LLMs with hundreds of billions of parameters is a non-trivial task that requires significant efforts in terms of both technical engineering and computational resources.  This paper represents the pioneering effort in exploring the use of exceptionally large language models as item encoders for the top-$N$ item recommendation task. It also provides the first confirmed evidence of the positive scaling effects of LLMs for item recommendation (from implicit feedback).
>
> Second, while both papers compare TCF with IDCF, this study arrives at a distinct  conclusion. [1] claimed that TCF can only compete with IDCF when the text encoders or LMs are jointly fine-tuned. However, this work discovered that when using a super large LLM, the frozen text encoder with pre-extracted offline representation can already be competitive with ID embeddings.
> This finding represents a significant progress, as fine-tuning very large LLMs  could require 100 or 1,000 times more computation, which is often impractical for large-scale real-world applications. Therefore, this is regarded as a remarkable finding that holds significant implications for recommender systems in terms of moving away from relying on explicit itemID embeddings.
>
> Beyond these contributions, authors have also studied the zero-shot transfer learning effects of TCF with super large LLMs pre-trained on a large-scale  dataset (Bili8M), albeit with surprisingly unexpected results. At last, they have benchmarked numerous well-known open-source LLMs for recommender systems, laying the foundation for future research.
>
> The results of this research not only validate some previous findings derived from smaller LMs but also uncover new insights beyond existing studies. These experimental outcomes, regardless of their positive or negative nature, call for further contemplation and discussion within the research community.  It is worth emphasizing that these discoveries were made possible through extensive and costly empirical studies, making a significant contribution to the existing literature.

---

### Decision · Program_Chairs · 2024-01-16

Reject